# Developmental variability channels mouse molar evolution

**Luke Hayden**[1,2], **Katerina Lochovska**[3], **Marie Sémon**[1], **Sabrina Renaud**[4], **Marie-Laure Delignette-Muller**[4], **Maurine Vilcot**[5], **Renata Peterkova**[6], **Maria Hovorakova**[7†*], **Sophie Pantalacci**[1*]

[1]Laboratoire de Biologie et Modélisation de la Cellule, Université de Lyon, CNRS UMR 5239, Ecole Normale Supérieure de Lyon, Université Claude Bernard Lyon1, INSERM U1210, Lyon, France; [2]Institut de Génomique Fonctionnelle de Lyon, Université de Lyon, CNRS UMR 5242, Ecole Normale Supérieure de Lyon, Université Claude Bernard Lyon 1, Lyon, France; [3]1st Department of Medicine, First Faculty of Medicine, Charles University and General University Hospital in Prague, Prague, Czech Republic; [4]Laboratoire de Biométrie et Biologie Evolutive, Université de Lyon, Université Claude Bernard Lyon 1, CNRS UMR 5558, VetAgro Sup, Villeurbanne, France; [5]Master de Biologie, École Normale Supérieure de Lyon, Université Claude Bernard Lyon I, Université de Lyon, Lyon, France; [6]Department of Histology and Embryology, Third Faculty of Medicine, Charles University, Prague, Czech Republic; [7]Department of Developmental Biology, Institute of Experimental Medicine, The Czech Academy of Sciences, Prague, Czech Republic

**\*For correspondence:**
Maria.Hovorakova@lf1.cuni.cz
(MH);
sophie.pantalacci@ens-lyon.fr (SP)

**Present address:** †Institute of Histology and Embryology, First Faculty of Medicine, Charles University in Prague, Prague, Czech Republic

**Competing interests:** The authors declare that no competing interests exist.

**Abstract** Do developmental systems preferentially produce certain types of variation that orient phenotypic evolution along preferred directions? At different scales, from the intra-population to the interspecific, the murine first upper molar shows repeated anterior elongation. Using a novel quantitative approach to compare the development of two mouse strains with short or long molars, we identified temporal, spatial and functional differences in tooth signaling center activity, that arise from differential tuning of the activation-inhibition mechanisms underlying tooth patterning. By tracing their fate, we could explain why only the upper first molar reacts via elongation of its anterior part. Despite a lack of genetic variation, individuals of the elongated strain varied in tooth length and the temporal dynamics of their signaling centers, highlighting the intrinsic instability of the upper molar developmental system. Collectively, these results reveal the variational properties of murine molar development that drive morphological evolution along a line of least resistance.

## Introduction

Evolutionary developmental biology postulates that developmental mechanisms confer specific variational properties on a trait, and can thereby channel its evolutionary trajectory. In extreme cases, a trait may repeatedly evolve similar phenotypes. Although this conceptual framework is central to evo-devo, it lacks cohesive supporting evidence. Only rarely the different levels of variation are bridged, from developmental variation to adult variation, and from variation between individuals to variation between populations or species. In this study focused on mouse molar teeth, we bridged these levels and reveal particularities of the developmental system that explain the morphological variation produced and its repeated appearance.

The idea that developmental mechanisms may channel and even direct the evolution of phenotypes is central to evo-devo (*Brakefield, 2006*; *Brakefield, 2011*; *Hendrikse et al., 2007*). It relies

**eLife digest** Over time species develop random mutations in their genetic sequence that causes their form to change. If this new form increases the survival of a species it will become favored through natural selection and is more likely to get passed on to future generations. But, the evolution of these new traits also depends on what happens during development.

Developmental mechanisms control how an embryo progresses from a single cell to an adult organism made of many cells. Mutations that alter these processes can influence the physical outcome of development, and cause a new trait to form. This means that if many different mutations alter development in a similar way, this can lead to the same physical change, making it 'easy' for a new trait to repeatedly occur. Most of the research has focused on finding the mutations that underlie repeated evolution, but rarely on identifying the role of the underlying developmental mechanisms.

To bridge this gap, Hayden et al. investigated how changes during development influence the shape and size of molar teeth in mice. In some wild species of mice, the front part of the first upper molar is longer than in other species. This elongation, which is repeatedly found in mice from different islands, likely came from developmental mechanisms.

Tooth development in mice has been well-studied in the laboratory, and Hayden et al. started by identifying two strains of laboratory mice that mimic the teeth seen in their wild cousins, one with elongated upper first molars and another with short ones. Comparing how these two strains of mice developed their elongated or short teeth revealed key differences in the embryonic structures that form the upper molar and cause it to elongate. Further work showed that variations in these embryonic structures can even cause mice that are genetically identical to have longer or shorter upper first molars.

These findings show how early differences during development can lead to small variations in form between adult species of mice. This study highlights how studying developmental differences as well as genetic sequences can further our understanding of how different species evolved.

on the concept that developmental mechanisms bias the direction and the amount of variation available to both natural selection and neutral drift. This was recognized early, mainly under the term of 'developmental constraints' (*Gould and Lewontin, 1979*; *Smith et al., 1985*) and studied from different viewpoints.

In the field of quantitative genetics, the analysis of phenotypic variation in crosses provides the direction of the genetic correlation between the different traits characterizing a shape. It was found that the direction of the genetic correlation between traits can match the direction of phenotypic variation within species, that itself matches the phenotypic variation between divergent populations or species. This suggested that phenotypic evolution happens along 'genetic lines of least resistance'(*Schluter, 1996*). Because the structure of the genetic correlations itself also match over long time spans (e.g. the G matrix was found to be similar among distant species), these 'genetic lines' are thought to be produced by developmental constraints, rather than by the persistence of specific genetic variants. This finding was recovered in a number of models including the molars of murine rodents (*Renaud and Auffray, 2013*; *Renaud et al., 2006*; *Renaud et al., 2011*).

The study of developmental systems in terms of their evolution also argues for a role of development in orienting morphological diversification (*Smith et al., 1985*; *Sears, 2014*), including in the tooth model. This is recognized under the more specific term of 'morphogenetic constrains'. The patterns of variation recovered following experimental perturbations of amphibian development (*Alberch and Gale, 1985*; *Oster et al., 1988*) or mouse molar development (*Kavanagh et al., 2007*) predicted the pattern of morphological variation seen among species. By experimentally manipulating the mouse tooth germ or tinkering with one or two parameters of a computational model of tooth morphogenesis, *Harjunmaa et al. (2014)* have reproduced evolutionary transitions seen in the fossil record, implying that the same construction rules have constrained morphogenesis since early mammals.

Despite this long interest and recent advances, there is still active discussion about how much development really influences evolutionary trajectories (*Laland et al., 2014*; *Smith et al., 1985*).

One difficulty is that the different levels of variation are rarely bridged in a single model: from variation in embryo to variation in adult, and from variation in populations to variation between well-diversified species. Mouse molars represent a rare opportunity to construct such a bridge: the *Mus* genus is well diversified, with many instances of repeated evolution and well-characterized trajectories of phenotypic variation in molar shape. Moreover, molar development is well known in the laboratory mouse.

Mice are part of the larger group of murine rodents (Old World mice and rats). In this group, the main direction of phenotypic variation in first molar shape divides species with narrower molars (with narrower cusps, for example dwarf mice of *Nannomys* subgenus in *Figure 1*) from species with broader molars (with broader, more roundish cusps, e.g. wood mice or grass mice of *Arvicanthis* genus, *Figure 1*). These differences in tooth morphology have been associated with different diet preferences, narrow teeth being mostly found in faunivorous rodents while broad teeth are characteristics of herbivorous ones, because of the latter allowing the consumption of harder and more abrasive food (*Gómez Cano et al., 2013*). Molar tooth morphology thus reflects adaptation in murine rodents as seen in many other mammals. However, whatever the mean morphology of a taxon, the variation within a taxon (e.g. house mouse and wood mouse), including at the population level, seems to reproduce, to a lesser degree, the basic variation ranging from narrow to broad tooth (*Renaud et al., 2009*; *Renaud et al., 2011*). Such micro-scale variation is more likely to be shaped by developmental properties, rather than adaptation. The high integration between the variation of lower and upper first molars suggests that both evolved in a concerted manner under similar developmental constraints. In summary, this alignment of the main phenotypic variation across the taxonomic scale suggests that murine molars evolve of along a line of least resistance, with adaptation occurring along the line imposed by developmental properties. On top of that, in some species or populations, only the upper molar tends to elongate, specifically from its anterior part which may even form an additional small cusp (*Misonne, 1969*; *Renaud et al., 2011*). This additional cusp is especially common in the *Mus* genus (yet occasionally seen in other murine species *Misonne, 1969*). For example, it is especially marked in some species of the *Mus (Nannomys)* subgenus, and also repeatedly seen in diverse house mouse populations (*Renaud et al., 2011*, see later *Figure 1*). In particular, it evolved independently in many *Mus (Mus) musculus domesticus* island populations (e.g. on several Corsican islands, Marion Island [*Renaud et al., 2011*], Orkney islands [*Ledevin et al., 2016*; *Renaud et al., 2018*], as well as on Kerguelen and the Canary islands, S. Renaud, [*Ledevin et al., 2016*]). Interestingly, anterior elongation is found associated with increased body size in domestic mouse populations (*Renaud et al., 2011*), being trapped on island (and following 'Foster's rule', or 'island rule' where small mammals become gigantic) or in cold environments. In conclusion, two intermingled developmental constraints seem to act here to channel phenotypic variation and evolution in murine rodents (or said differently, to shape a line of least resistance): one acts on both first molars, and the other one acts on the upper molar only, favoring its repeated anterior elongation in many independent populations and species.

Molar development is well known (*Balic and Thesleff, 2015*; *Peterkova et al., 2014*), so that in a previous study, we had put forward some hypothesis for the developmental basis of this line of least resistance characteristic of the upper molar (*Renaud et al., 2011*). During molar development, signaling centers, called 'enamel knots', are positioned in the epithelium by activation-inhibition mechanisms, and determine the location where the crown, or later the cusps, form *Jernvall et al. (1994)*; *Sadier et al. (2019)*. Although murine rodents lack premolar teeth, structures found transiently in mouse embryos are thought to correspond to their rudiments, each with their own signaling centers, notably secreting the signaling molecule Shh (*Lesot et al., 1998*; *Peterková et al., 1996*; *Prochazka et al., 2010*; *Viriot et al., 2000*; *Peterková et al., 2002*). Following molar row initiation around 11.5 dpc, the dental epithelium progressively invaginates into the mesenchyme along the antero-posterior jaw axis, and Shh-signaling centers are patterned sequentially in this epithelial ridge (*Prochazka et al., 2010*; *Sadier et al., 2019*). In mandible, the first and most anterior signaling center, associated with a discrete epithelial swelling, has formed by 12.5 dpc but disappears soon after (in CD1 mice). A second signaling center (R2 signaling center) has formed more posteriorly by 13.5 dpc, where the epithelial ridge locally enlarges and gives rise to a more prominent bud called R2. This structure was interpreted as a rudiment of a suppressed premolar (*Prochazka et al., 2010*; *Viriot et al., 2000*), and can form a small tooth in mutant conditions interpreted as an atavistic premolar (*Cobourne and Sharpe, 2010*; *Lagronova-Churava et al., 2013*; *Lochovska et al., 2015*;

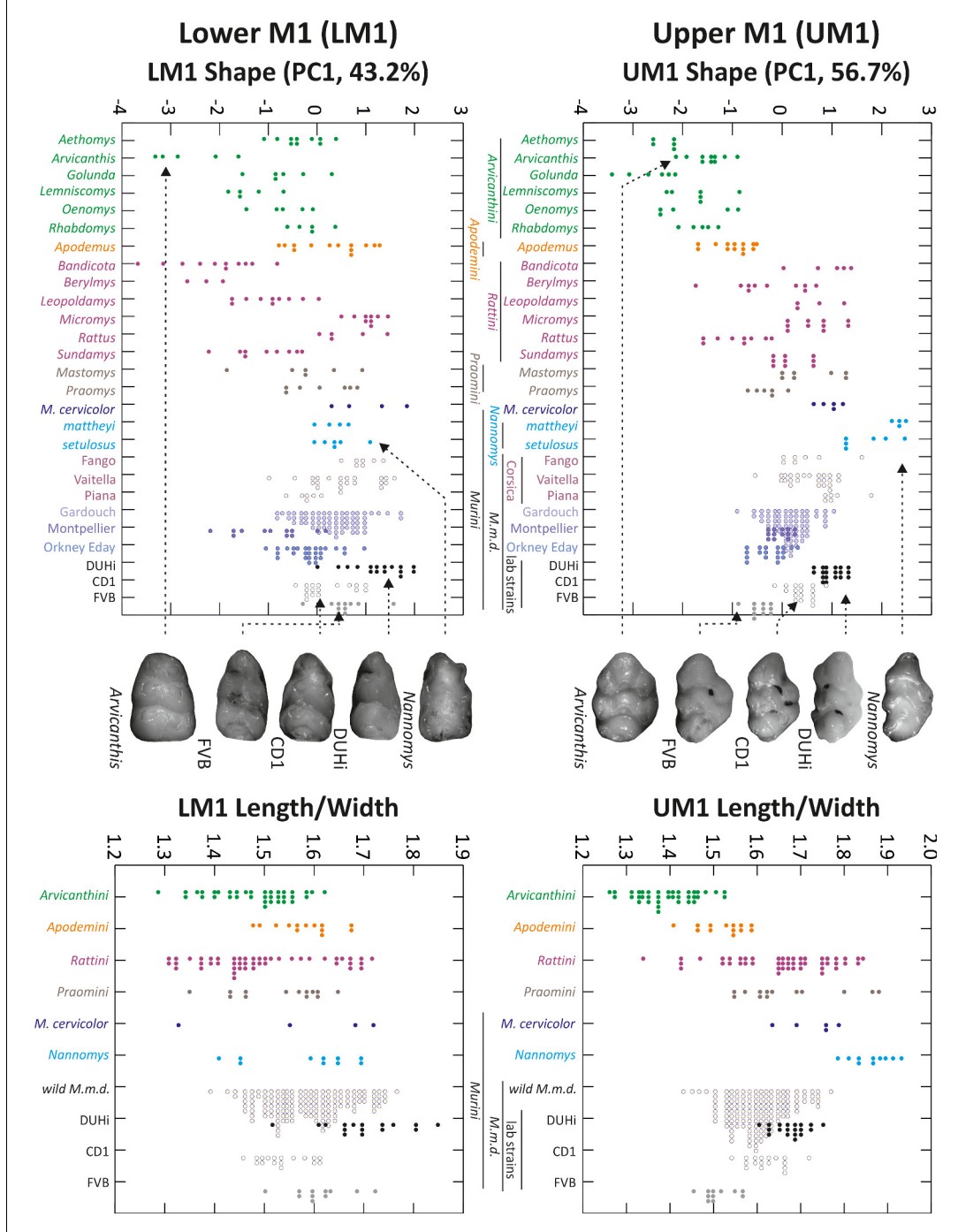

**Figure 1.** Morphological variation in murine first molar shape, based on 2D outlines. Morphological variation of the first molar, both upper and lower, examined in individuals from several murine species, including mouse natural populations of the house mouse *Mus musculus domesticus* (M.m.d) and three mouse strains (*Mus musculus*). The left panel shows the first axis of a principal component analysis of 2D outline descriptors (Fourier coefficients). The right panel shows the length/width ratio of the molar (measures were taken on the same individuals as in the left panel, but they were grouped according to the phylogenetic groups shown in the upper left panel). Tooth images in the middle represent an example of the strain or species indicated, with arrows from each pointing to the point for that individual on each graph. UM1 – upper first molar; LM1 – lower first molar.

The online version of this article includes the following figure supplement(s) for figure 1:

**Figure supplement 1.** Representative examples of adult upper molar morphology in DUHi and FVB.

*Peterková et al., 2005*). This signaling center also soon vanishes and by 14.0 dpc, a third, even more posterior, signaling center has formed, called the early M1 signaling center. In the lower jaw, the R2 signaling center interacts in a complex manner with the signaling center of the first molar. Both transiently co-exist before fusing to form the mature M1 signaling center known as the pEK (*Lochovska et al., 2015*; *Sadier et al., 2019*). This pEK drives 'cap transition', a morphogenetic transition during which the epithelium starts to fold around the mesenchyme to form the tooth crown. From an early cap stage, the R2 primordium becomes thus integrated into the anterior part of the first lower first molar.

Although a similar R2 bud is also present in the upper jaw (*Lesot et al., 1998*; *Peterková et al., 1996*), its developmental relationship to the upper first molar is less clear. It is not incorporated during the initial stage of molar cap development, as seen for its lower counterpart (*Peterkova et al., 2006*). Recently, we have shown that the distance between R2 and M1 signaling centers is larger in the upper jaw, where the two signaling centers do not fuse at the cap stage (*Sadier et al., 2019*). The anterior position of R2 and the difference between lower and upper R2, make these structures excellent candidates to explain that the upper molar, and not its lower counterpart, evolves so repeatedly towards anterior elongation (*Renaud et al., 2011*).

To get insights into the developmental basis of upper first molar repeated evolution, we looked for two mouse strains reflecting the above-mentioned evolutionary variations: a 'broad and upper-short' versus a 'narrow and upper-elongated' strain. It was noted long ago that two laboratory mouse strains displayed the elongated upper molar morphology with a small additional anteriormost cusp (C3H, 101 strain from Harwel, *Grüneberg, 1965*). In our effort to test the correlation between this morphology and a large body size, we found that the DUHi mice, an inbred strain that was established following artificial selection for increased body size (*Bünger and Herrendörfer, 1994*; *Bünger et al., 1982*), display narrow molars with elongated first upper molars, and an additional cusp in some individuals (*Figure 1—figure supplement 1*). In contrast, the FVB mice, an inbred strain often used to maintain genetic modifications, display wide molars with short first upper molars (*Figure 1—figure supplement 1*). After checking that these strains mirror mouse molar phenotypic evolution, we looked for developmental variation in the dynamics of R2 rudimentary buds between strains and jaws, but also within strains. We asked whether variational properties of the upper molar developmental system, whether qualitative or quantitative, may predict the evolutionary variation of the first upper molar.

## Results

### The variation between DUHi and FVB molar morphology follows the murine evolutionary lines of least resistance

In order to place these two strains within the context of natural variation in molar tooth shape, we compared the outline of first molars in a number of murine groups, including *Mus musculus domesticus* from the wild and three lab strains (two inbred strains: FVB, DUHi; one outbred strain CD1). We performed a principal component analysis (PCA) of outline descriptors (obtained from an outline analysis of the 2D outline, see methods) as an agnostic method (i.e. it does not make strong assumptions about the shape of the underlying variation) to reveal the main direction of variation in the dataset, and a direct comparison of molar Length/Width ratio (*Figure 1*). For both the upper and lower molar, the first axis of the PCA contrasts broad with narrow outlines. Hence, this can be considered to be the most important aspect of the outline variation for both teeth, with a morphology ranging from short, compact and rounded teeth to long and narrow teeth. However, this variation is more pronounced for the upper molar (PCA UM1 = 62% instead of PC1 LM1 = 46% of variance) because it also involves a change focused at the anterior part of the tooth, and opposing short *vs.* anteriorly elongated UM1. This variation corresponds to the evolutionary trend seen repeatedly for the upper molar in the *Mus* genus: the anterior elongation that can even take the form of a small additional cusp (*Renaud et al., 2011*; *Stoetzel et al., 2013*). We note that although the PCA of the outline descriptors could in principle separate these two types of variation, this was not the case, showing that these two types of variation are correlated in the upper molar (i.e. anterior elongation occurs in otherwise narrow molars, while broad molars have a short anterior part).

The position of FVB and DUHi strains along the PC1 axis and length/width ratio indicate that the direction of variation between the two strains recapitulates the direction of variation seen in murine rodents as a whole (e.g. broader FVB molars versus narrower DUHi molars, lower and upper M1). Moreover, it recapitulates the variation specifically seen in the upper first molars of the *Mus* genus (short FVB versus anteriorly elongated DUHi upper molars, with additional cusp). Indeed, the two strains are representative of extreme wild *Mus musculus domesticus* samples: DUHi teeth fall with the most elongated upper molars (e.g. samples from a very small Corsican island: Piana), while the FVB upper molars fall with the most 'short and wide' upper molars (e.g. samples from the continent (Gardouch locality) or from a large Orkney island: Eday). This validates the choice of these strains, as recapitulating the two intermingled main phenotypic variations seen in murine rodents. The next step was to examine the developmental basis of interstrain variation in lower and upper molars.

## Modeling a numeric embryonic age to account for inter-strain timing differences and the dynamic nature of the system

In order to allow the developmental trajectories of the developing molar teeth to be compared, embryonic samples were taken from a wide developmental window in the two strains in question (12.0 to 15.5 dpc). For each strain, multiple litters were sampled every half day, ensuring even coverage due to the slight variation in developmental stage within litters. Taking account of the dynamic nature of the system required a numeric estimation of the embryonic age of embryos (i.e. an age reflecting the progress in embryogenesis). Neither the age in days post-coïtum nor the embryonic body weight solely provide a correct estimation (for further explanation, see Materials and methods and Appendix 1). Therefore, in this study, we devised a Bayesian modeling approach to compute a strain-specific embryonic age for each embryo (later called $_c$dpc) by combining these two information (see Materials and methods and Appendix 1 for further precisions). Such a model learns from the data, and could also take into account strain differences without imposing a priori assumptions. Because the model does not comprise information on actual stage differences and we estimate $_c$dpc for each strain, it will not correct for developmental stages differences between strains, but instead reveal them by providing a time framework, the computed age (e.g. cap transition occurs at earlier $_c$dpc in FVB). The detailed model and the script are provided in Appendix 1. An important stochastic term in the model corresponds to the part of the inter litter variation of the body weight due to pregnancy, for which we explored two values (Figures of Appendix 1). The first one, which we consider realistic, corresponds to a maximum effect on weight of 20 mg for a 200 mg embryo. The second one, which we consider an extreme upper bound, corresponds to a maximum effect on weight twice as important (40 mg for a 200 mg embryo). The computed embryonic ages (next called $_c$dpc, for computed days post coïtum) presented in the main text have been estimated using the realistic parameter, but the results are qualitatively robust using the upper bound parameter (*Appendix 1— figures 2–8*). This demonstrates that our results are robust to noise in embryonic age estimation. We also used embryonic age directly estimated from body weight (a simplification of the previous model, similar to what we used in our previous studies, *Pantalacci et al., 2009*). All the results shown in this study were robust in relation to these estimations, although they differ slightly (*Appendix 1— figures 2–8*). In the next paragraphs, the results obtained for upper molar are mostly presented in the main figures and results for lower molar will be found in Appendix 1.

## The range of possible developmental states and the developmental trajectory taken differs between strains, especially for the upper molar

We proceeded to compare the DUHi and FVB lower and upper developmental systems. In both jaws and strains, we see *Shh* expression at the signaling center of the R2 bud (*Figure 2* and *Figure 2— figure supplement 1A–D*). This expression then fades away (*Figure 2—figure supplements 1* and *2* E-H). A second spot of expression appears, which represents the early M1 signaling center (*Figure 2* and *Figure 2—figure supplement 1I–L*). As development proceeds to the cap transition stage, the M1 expression zone increases in size (*Figure 2* and *Figure 2—figure supplements 1– 2*) to form a 'mature M1 signaling center' (differences between lower and upper jaw in this process will be emphasized later), and the tooth continues to develop (*Figure 2—figure supplements 2– 3*). A simple examination of embryonic series suggested differences in the dynamics of the signaling centers (*Figure 2*, e.g. see I-J versus K-L). In the upper jaw of DUHi mice, we frequently see the co-

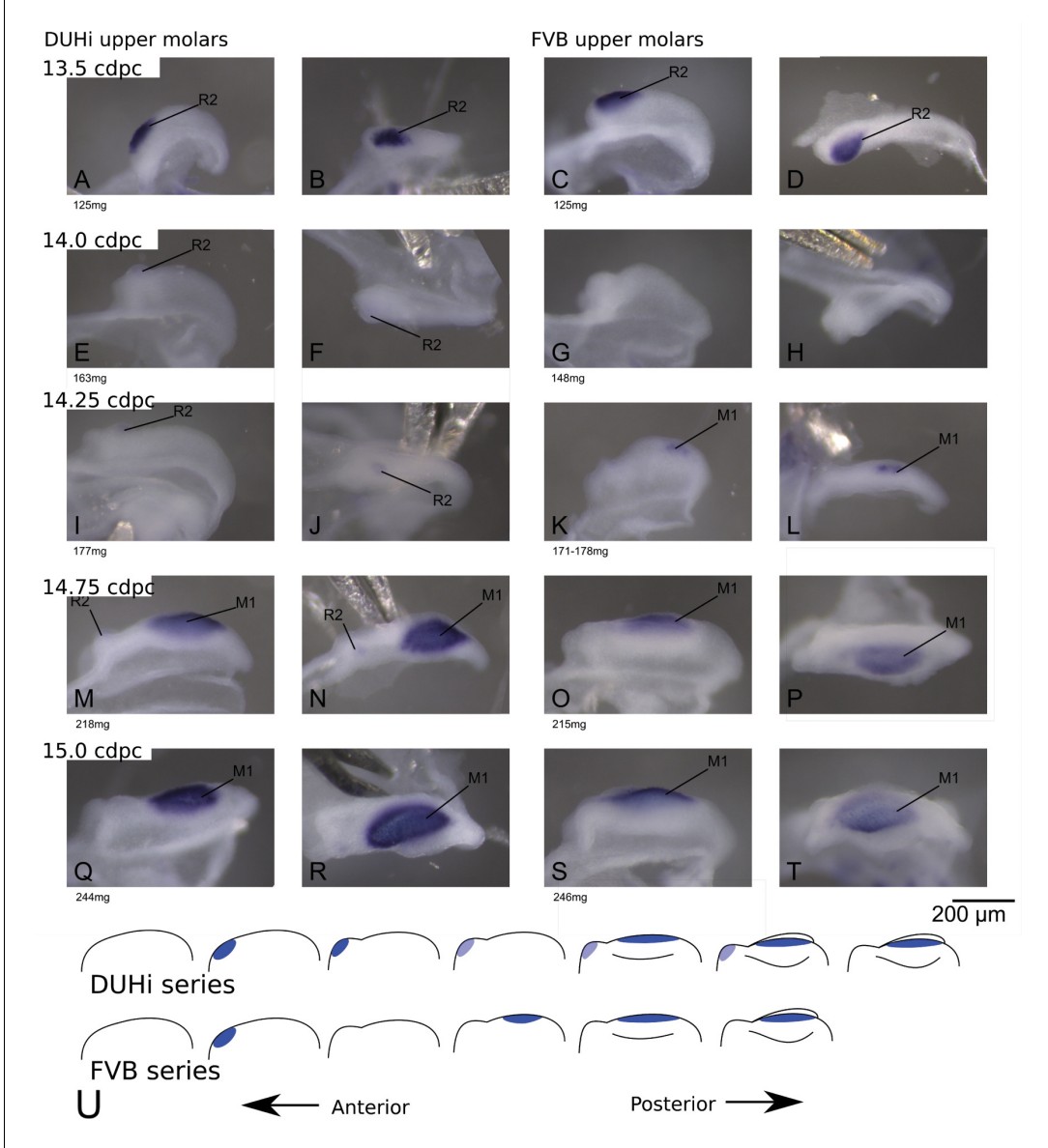

**Figure 2.** Comparative early molar tooth development in upper molar epithelia of DUHi and FVB mice. Dissociated upper molar dental epithelia epithelia of DUHi (left-most two columns) and FVB embryos (right-most two columns), marked for *Shh* expression with in situ hybridisation. Samples represent a developmental series of early molar development, corresponding to 12.5dpc to 15.0dpc in FVB and 13.0dpc to 15.5dpc in DUHi. Embryo weight is noted below each sample and its equivalent in computed embryonic age is noted above each row of samples. Two images of each sample, a side and a top view, are shown.

The online version of this article includes the following figure supplement(s) for figure 2:

**Figure supplement 1.** Comparative early molar tooth development in lower molar epithelia of DUHi and FVB mice.
**Figure supplement 2.** Comparative late molar tooth development in upper molar epithelia of DUHi and FVB mice.
**Figure supplement 3.** Comparative late molar tooth development in lower molar epithelia of DUHi and FVB mice.

occurrence of the fading R2 spot with a distinct M1 spot (*Figure 2M–T* and *Figure 2—figure supplements 1–2* M-T).

To allow comparisons between jaws and strains, we proceeded to examine development in a quantitative fashion (*Figure 3—figure supplements 1–2*). For each sample dated in cdpc, four key characters were scored for two to three possible states (characters were: the expression of *Shh* at the premolar R2 signaling center, the expression of *Shh* at the M1 signaling center, the progression of the bud-cap transition at the level of M1, and the protrusion of the dental epithelium at the level

of R2, *Supplementary file 1*). Then, we combined these four characters to compute the developmental state of each sample (*Figure 3* and *Figure 3—figure supplement 3*). Mathematically speaking, this yields 54 possible total states of the dental epithelium, which can be conceptualized as a theoretical 'developmental space' through which each individual will move as it develops. However, not all of this space is occupied; the number of states observed was much smaller than the mathematical maximum theoretically possible (11 states for the upper molar, *Figure 3*; and 9 states for the lower molar, *Figure 3—figure supplement 3*). This is because the unfolding of normal development imposes rules on characters' state transitions and combinations. For example, transitioning from a well-developed cap to a bud seems impossible, because normal tooth development proceeds from bud to cap stage. Yet it remains conceivable in mutant situations, where morphogenesis would be

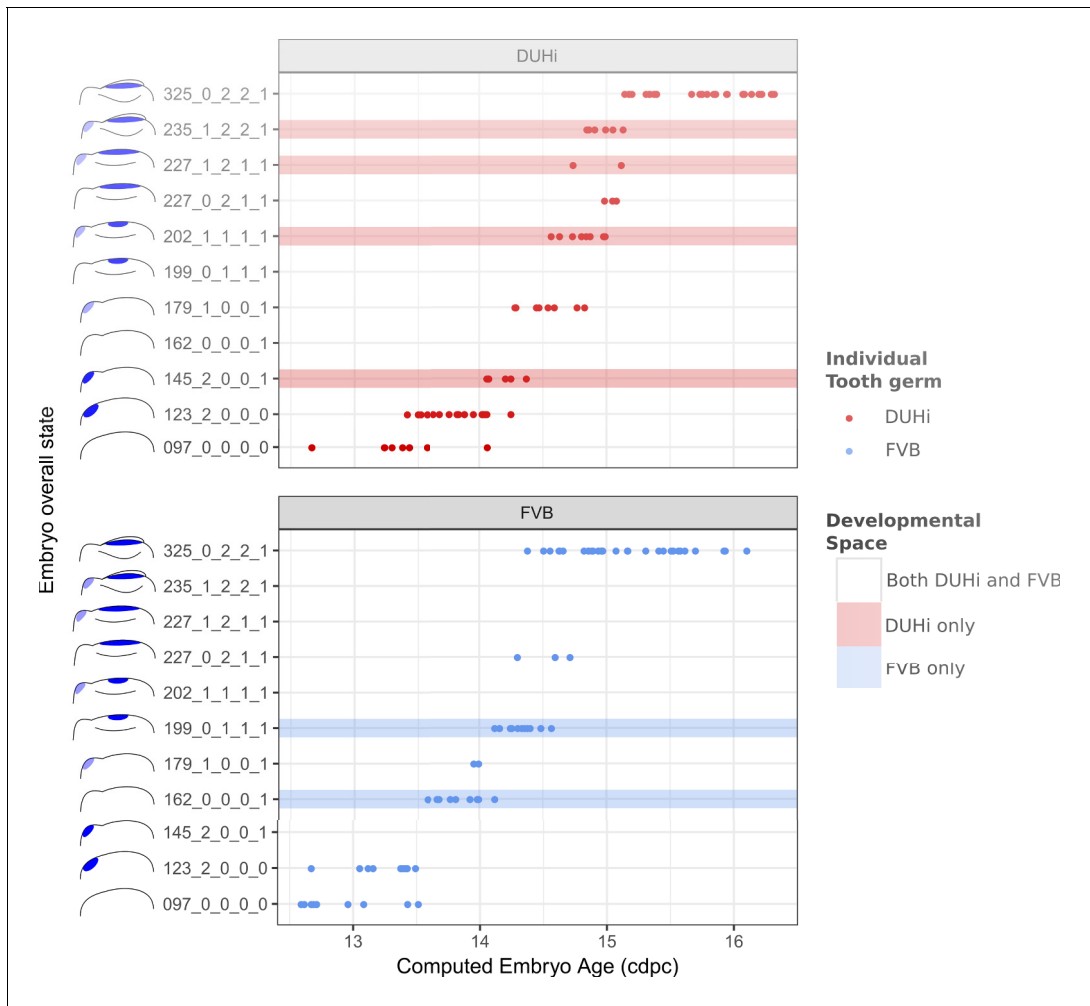

**Figure 3.** The range of possible developmental states differs between FVB and DUHi developing upper molars. Temporal distribution of developmental state of the developing upper molar, produced by combining a value for each of the four scores for a given sample, based on criteria from *Supplementary file 1*. Each of the developmental states observed are schematized by a cartoon and ordered according to the average embryonic weight of the samples within that group. They are named with this weight, followed by the value of the four scores (eg. 097_0_0_0_0, means average weight 97 mg, 0 value for the four scores: R2 Shh expression, M1 Shh expression, cap transition state, R2 protuberance state). Exclusive developmental states are highlighted according to whether they are found in DUHi only (red) or in FVB only (blue). The temporal axis is given by computed embryo age ($_c$dpc). The dental epithelium is oriented with anterior part to the left. Scale bar = 200 μm. Developmental progression is summarized in schematic form below (U).

The online version of this article includes the following figure supplement(s) for figure 3:

**Figure supplement 1.** Comparative progression of scored characteristics in upper dental epithelium.

**Figure supplement 2.** Comparative progression of scored characteristics in lower dental epithelium.

**Figure supplement 3.** The range of possible developmental states differs between FVB and DUHi developing lower molars.

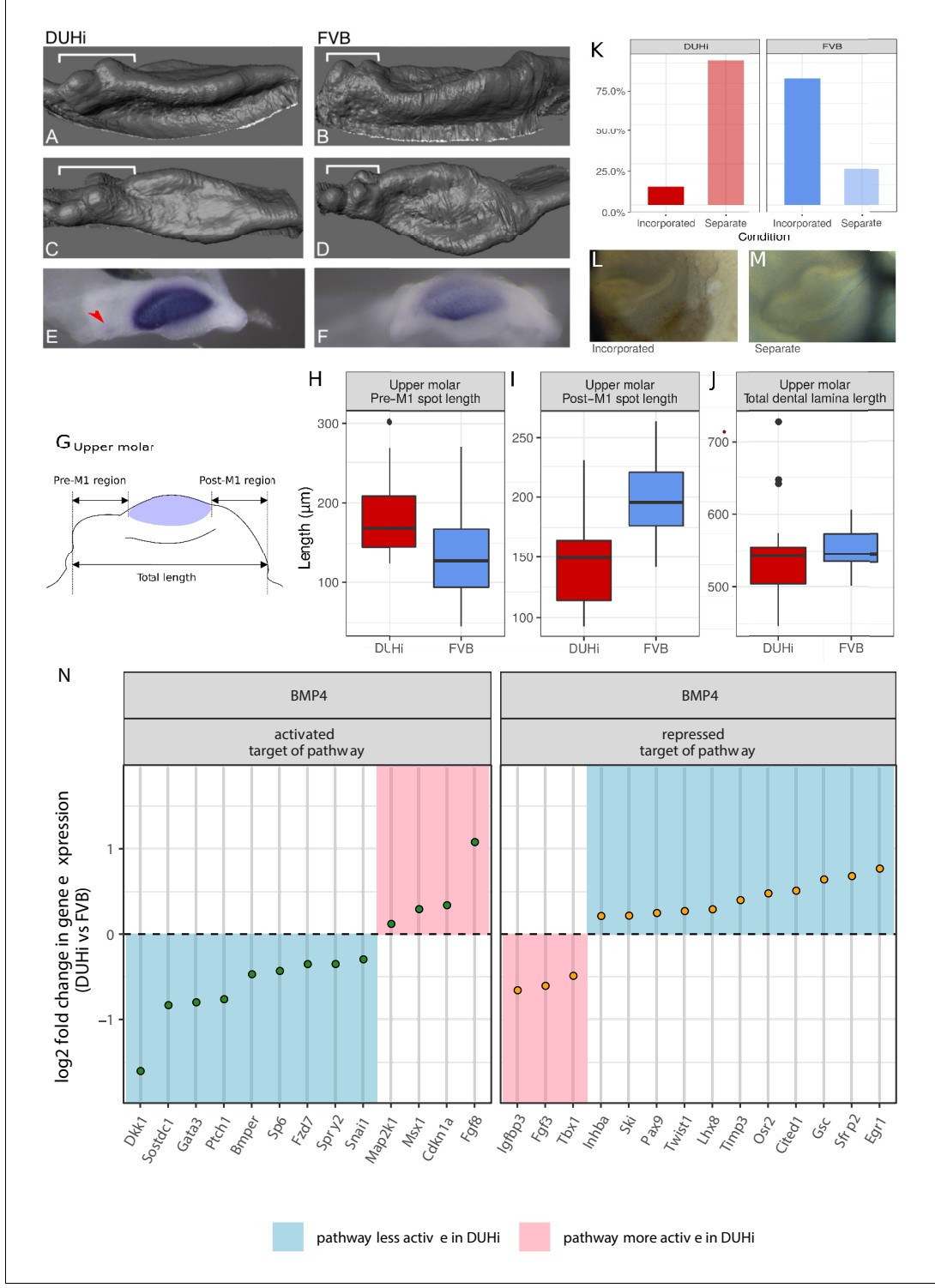

**Figure 4.** FVB and DUHi strains differ in the balance of activation-inhibition mechanisms. (**A–F**) Comparative 3D morphology of the epithelial part of developing upper molars in DUHi (**A, C**) and FVB (**B, D**) strains at cap transition. A, B are side views and C, D upper views. The variable anterior region including R2 rudimentary bud is denoted by the bracket. Expression of *Shh* at the same timepoint is shown below for DUHi (**E**) and FVB (**F**), for comparative purposes. The red arrowhead in E points to R2 signaling center. In F, the faint staining anterior to the mature M1 signaling center might correspond to re-expression of Shh in cells that formed R2 signaling center. (**G–J**) panel G shows the three measurements taken from all epithelial samples of a computed age of 14.25–15 cpdc (between 180 and 250 mg weight), comparing DUHi with FVB samples. The measures were taken between the

*Figure 4 continued on next page*

*Figure 4 continued*

anterior or posterior limit of dental epithelium and the anterior or posterior limit of the M1 signaling center, as shown on the cartoon. Boxplots H-J show the results in upper molars (see *Figure 4—figure supplement 1* for lower molars). Pre-M1 region and Post-M1 regions are significantly different in DUHi *versus* FVB mice (t-test; p<0.01, see *Supplementary file 4*). (K–M) The developing molars of DUHi and FVB mice react differently when cultured in vitro at 13.0 dpc (p=0.015 in an exact Fisher test, see *Supplementary file 4*): R2 bud tends to form a clear individualized bud (M) in most DUHi tooth cultures (n = 18), whereas a single developing tooth (L) is seen in most FVB tooth cultures (n = 18). (N) Target genes of the Bmp4 pathway differentially expressed between the two strains at the cap transition. Differential expression analysis was performed on both lower and upper molar samples, taking molar type into account in the statistical treatment by DEseq2. Genes were classified as targets activated or repressed by the pathway based on *O'Connell et al. (2012)*. The log2 fold change in DUHi as compared to FVB is shown (positive: expression level increased in DUHi tooth germs; negative: expression level decreased in DUHi tooth germs). Depending if the gene is an activated or a repressed target, and is increased or decreased in DUHi tooth germs, it may suggest that the pathway is more active (pink) or less active (blue) in DUHi tooth germs.

The online version of this article includes the following figure supplement(s) for figure 4:

**Figure supplement 1.** Comparative measurements in lower molar dental epithelium.
**Figure supplement 2.** Comparative transcriptomics of DUHi versus FVB cap stage tooth germs.

---

stopped and the tissue would start regress. Combining no protuberance in R2 zone with a cap transition is unexpected, because R2 develops before M1. Combining large M1 *Shh* expression and no cap transition is also unexpected, because this large expression marks the presence of a PEK and PEK drives cap transition. Yet, again in mutant mice, things may happen differently, because the normal rules of development may be broken (e.g. defective PEK despite normal *Shh* expression). This emphasizes that the actual occupied proportion of developmental space and the trajectories through this space are properties of a given genotype with a given development.

A comparison of the distribution of developmental states seen in DUHi and FVB upper molar samples reveals that certain states are only present in one of these two strains. This is especially true for upper molar (3 DUHi-exclusive states, 2 FVB-exclusive states) as compared with lower molar (1 DUHi-exclusive state, evidenced by only two samples). Indeed, within a given weight range (from 14.25 to 14.75 $_c$dpc; 175 mg to 225 mg), all DUHi samples display a 'DUHi-exclusive' state, while all FVB samples display an 'FVB-exclusive' state. This corresponded to the period between the disappearance of the R2 signaling center and the maturation of the M1 signaling center. Therefore, the developmental differences between these two strains can be conceived of as each of the two strains following distinct trajectories through 'developmental space'. Moreover, these differences are especially marked in the upper molar.

## The premolar signaling center persists longer in DUHi than in FVB upper jaws

We then compared the timing of developmental events between the two strains. In upper molars (*Figure 3* and *Figure 3—figure supplement 1*), the R2 expression zone persists longer in DUHi than in FVB (logistic regression: presence of R2 signaling center activity in relation to computed embryo age, p<1.10E-8, see Supplementary statistical details 1). The R2 and M1 signaling center frequently co-occur in DUHi, and co-occurence of R2 and a mature (large) M1 signaling center was only seen in DUHi upper molars. In contrast, early M1 signaling center was seen without a R2 signaling center only in FVB mice. A logistic regression also revealed a statistically significant difference in the timing of the appearance of the M1 spot, which appears later in DUHi than in FVB embryos (p<0.01, see *Supplementary file 4*). The longer persistence of R2 signaling center and later appearance of the early M1 signaling center in the upper jaw of DUHi mice were associated with a more prominent R2 bud, as seen in dissociated epithelium (*Figure 2*, *Figure 4*-F) but also in 3D reconstructions of tooth germs (*Figure 4A–D*).

In the lower jaw, we observed a similar tendency, with R2 and M1 signaling centers occasionally coexisting only in DUHi, but never in FVB (*Figure 3—figure supplements 2–3*). In both strains, the lower R2 signaling center was ultimately part of the mature M1 signaling center known as the pEK

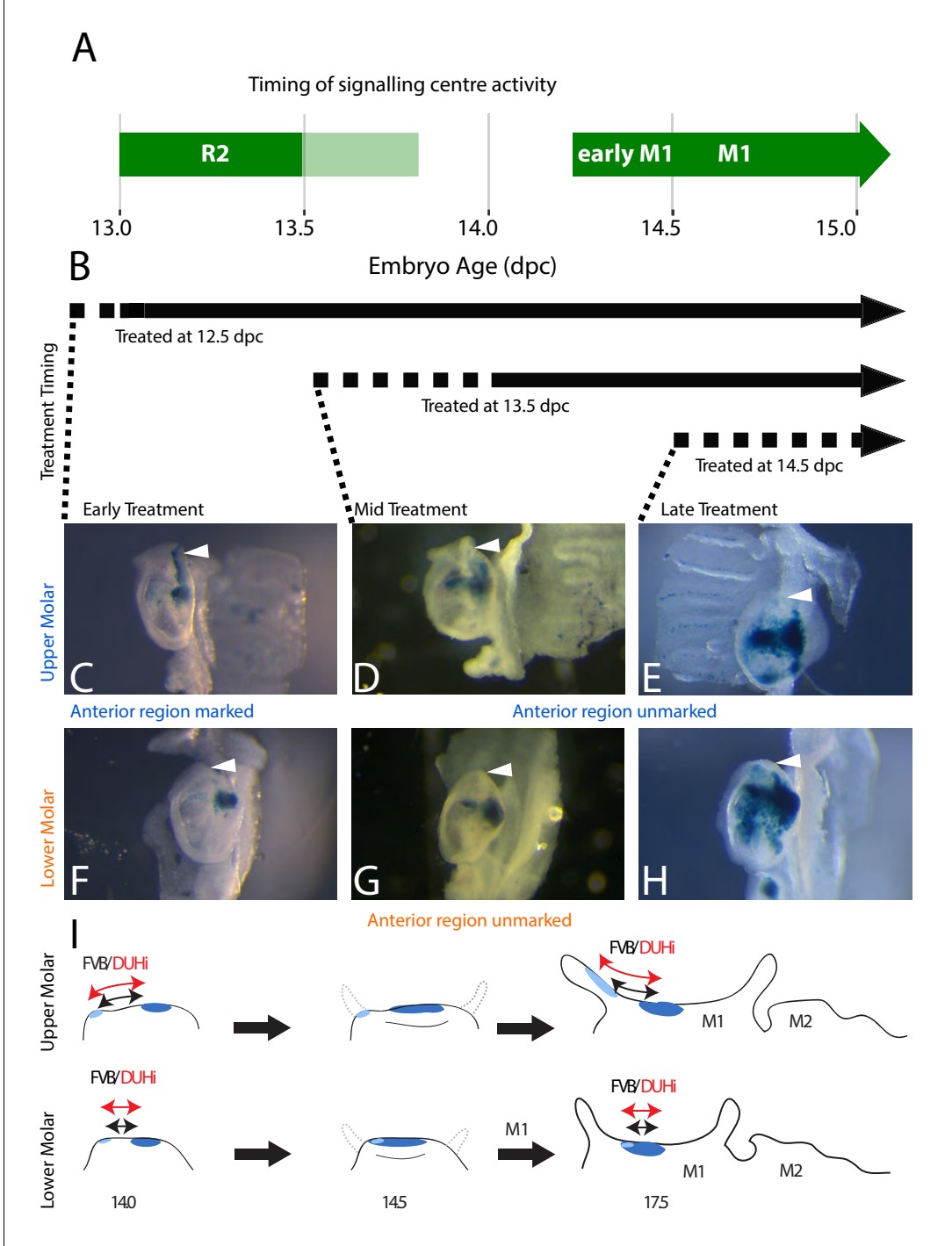

**Figure 5.** The contribution of R2 and M1 signaling centers to the anterior region of the upper and lower molar. (**A**) The timing of Shh expression in R2 and M1 signaling centers is shown in green (light green: faint *Shh* expression in R2). (**B**) In this context, a tamoxifen-inducible Shh-Cre^ERT line was used to induce the marking of *Shh*-expressing cells with β-galactosidase expression following tamoxifen injection. Time period of marking is indicated with black arrows, taking into account the roughly 12 hr-delay for activation (dashed line) plus the persistence of tamoxifen after injection. Early treatment (12.5 dpc injection) corresponds to activity of both R2 and M1 signaling centers, and will mark the progeny of both signaling centers during this period. Mid and late treatment (13.5dpc or 14.5dpc injections) correspond to activity of M1 signaling center only, and will mark exclusively the progeny of M1 signaling center (*Supplementary file 2*). (**C–H**) X-gal-stained epithelia of upper (**C–E**) or lower (**F–H**) molars at 17.5dpc are seen below, with the anterior region of the tooth (white arrowhead) marked in correspondence with the timing of the tamoxifen treatment. The presence of staining in the anterior part of the tooth in 12.5 dpc-treated individuals (**C**) with the lack of staining in later-treated individuals (**D** and **E**) indicates that R2 signaling center contributes to the anterior part of the first upper molar. The scheme in I summarizes results for lineage tracing with induction at 12.5 dpc, marking both

*Figure 5 continued on next page*

*Figure 5 continued*

R2 signaling center (light blue) and M1 signaling center (dark blue) descendant cells. Following M1 cap transition at 14.5 dpc, the tooth will develop anteriorly and posteriorly (shown on the 14.5 dpc scheme with dashed grey line). Only the upper R2 signaling center descendants are involved in anterior cervical loop formation. Differences in size and R2-M1 distance seen between FVB and DUHi strain (black *versus* red arrows) will preferentially impact the anterior part of the tooth in the upper molar only.

(*Figure 2—figure supplement 1R and P*), as shown for other mouse strains (*Lochovska et al., 2015*).

## FVB and DUHi strains differ in the balance of activation-inhibition mechanisms

We recently proposed that activation-inhibition mechanisms acting in a posteriorly growing domain rules the patterning and fate of the R2 signaling center (*Sadier et al., 2019*). The finding that R2 is larger and longer lived in DUHi mice suggests that the mechanism proposed in our previous study differs between the two strains. This could be at different levels, from posterior growth, rate of maturation, influence of mesenchyme or activation-inhibition mechanisms per se.

A prediction is that the positioning of the M1 signaling center should differ between the two strains. Our measurements revealed that the M1 signaling center was shifted posteriorly in DUHi mice: the pre-M1 signaling center region was longer in DUHi than in FVB samples (*Figure 4H, t*-test, p<0.01), while the post-M1 signaling center region was shorter in DUHi upper molars than in FVB (*Figure 4I, t*-test, p<0.01; see *Supplementary file 4*). No statistically significant differences in total dental epithelium length was found (*Figure 4J*). In the case of the lower molars (*Figure 4—figure supplement 1*, see *Supplementary file 4*), no statistically significant inter-strain differences emerged.

This finding suggests that the posterior growth rate is unchanged between the two strains. According to the sensitivity analysis performed in our previous study, two other parameters on top of posterior growth rate (Figure S3a in *Sadier et al., 2019* Supplementary Material) could produce more distant signaling centers (and possibly R2 rescue): one modulates activation-inhibition per se (Figure 5 and S2a in *Sadier et al., 2019*) and the other modulates the rate of production of the

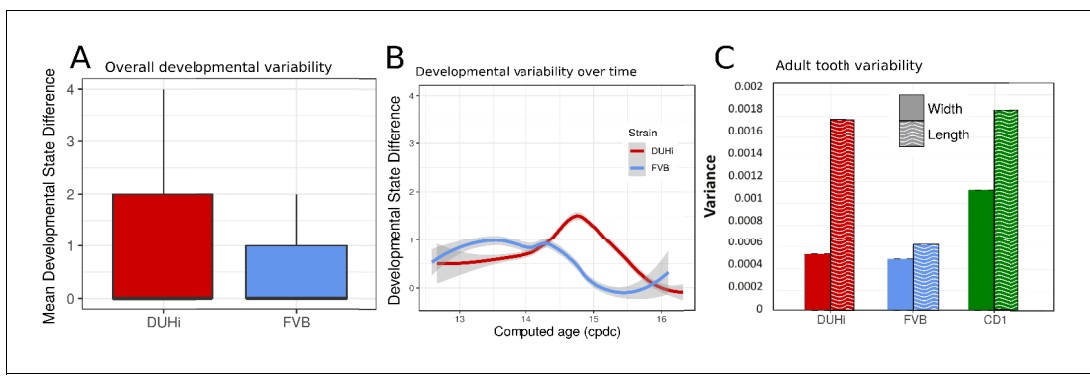

**Figure 6.** Developmental and adult variation is higher in DUHi first upper molars. (**A**) A measure of developmental variation in the developing upper molars of FVB and DUHi strains. The figure shows a boxplot of developmental state differences calculated for pairs of samples with less than 0.25 difference in computed embryonic age ($_c$dpc). Samples close in age are significantly more different in developmental state in DUHi versus FVB mice, according to a Wilcoxon test (p<0.001). See the Materials and methods for further explanation on this measure of developmental variation. (**B**) The mean developmental state difference between nearby samples (computed embryonic age difference <0.25) is plotted as the local regression line for both strains (standard deviation shown in grey). (**C**) Morphological variation in adult first molar, measured as the variance in molar width and length. Variation in length (but not width) is much greater in DUHi than that in FVB (both are inbred strains; p=0.095), and comparable to the outbred CD1 strain.

The online version of this article includes the following figure supplement(s) for figure 6:

**Figure supplement 1.** Developmental and adult variation in FVB and DUHi first lower molars.

mesenchyme signal that primes the tissue for activation-inhibition (Figure S4a in *Sadier et al., 2019*). Unfortunately, it is very unclear so far which molecular pathways would contribute to each of these two parameters. Despite that, we know many pathways that contribute positively or negatively to molar formation (inc Wnt, BMP, FGF, Activin, Edar pathways). In addition, some genes are known to suppress the potential of R2 to form a premolar-like tooth (e.g. *Klein et al., 2006*; *Peterkova et al., 2009*; *Sadier et al., 2019*). Next, we looked for gene expression differences between the two strains that would be consistent with a difference in these known pathways.

For that, we generated transcriptomes of lower and upper molar germs of FVB *versus* DUHi mice at the time when R2 and M1 signaling center co-exist in DUHi mice. We found a large number of differentially expressed genes (see Appendix 2 for detailed results, data available in *Supplementary file 2*). Among all those DE genes, we identified two genes, Spry2 and Sostdc1 (also known as Ectodin), whose knock-out causes the formation of a premolar-like tooth (*Ahn et al., 2010*; *Cho et al., 2011*; *Klein et al., 2006*). Where it was specifically investigated, this premolar tooth was demonstrated to arise from R2 revival (*Ahn et al., 2010*; *Klein et al., 2006*; *Peterkova et al., 2009*). The downregulation of these two genes in DUHi lower and upper molar samples (*Figure 4—figure supplement 2*) could thus help the partial rescue of R2 bud in this strain. We then focused on the Wnt and BMP4 pathways that have been shown to be key for tooth formation (*O'Connell et al., 2012*). Because tooth formation involves many genes with complex regulatory feedbacks within and between these two key pathways (and other pathways), we did not expect a change in activation-inhibition mechanisms to shift all target genes in a consistent direction. Rather, we expected to find a different equilibrium, with genes changed in both directions, but that may collectively indicate greater or instead weaker activation of these pathways in the DUHi mice. For the BMP4 pathway, there were 20 genes in favor of weaker BMP4 activation in DUHi (e.g. summing activated targets that are upregulated with repressed targets that are downregulated, see Appendix 2 for more detail and *Figure 4N*) *versus* only 7 genes in favor of greater BMP4 activity (e.g. summing activated targets that are downregulated with repressed targets that are upregulated). This two-fold difference is significant (p=0.01 in a $\chi^2$ test). For the Wnt pathway, we found no trend with 23 genes in favor of weaker Wnt activity in DUHi and 20 genes in favor of greater Wnt activity (*Figure 4N*). Intriguingly, this included 4 feedback inhibitors of the Wnt pathway upregulated in DUHi mice (Axin2, Kremen1, Osr2; Sfrp2) and 3 feedback inhibitors downregulated in DUHi (Dkk1, Wif1, Sostdc1). Finding these major regulators of Wnt activity in tooth development differentially expressed suggests that Wnt activity differs between FVB and DUHi, although we cannot orient it as for the BMP pathway. This is consistent with recent findings suggesting that both activation and inhibition of the Wnt pathway is required to make teeth, and Wnt activation needs to be carefully controlled by feedback mechanisms (including a crosstalk with the BMP4 pathway) to enable the sequential formation of teeth (*Järvinen et al., 2018*).

In conclusion, transcriptomic data further suggest a difference in the activation-inhibition balance between the two strains. The balance tends toward weaker BMP4 activity in DUHi mice at the cap transition stage, suggestive of lowered levels of activation in these mice (either directly, if BMP4 is part of activation-inhibition mechanisms per se, or indirectly, as part of the mesenchymal signal that enables activation-inhibition).

To further establish that these strains differ in activation-inhibition balance, we tested another prediction, that the two strains should react differently to the same perturbation of activation-inhibition mechanisms. Simply dissecting and culturing teeth ex vivo is known to provide such a perturbation, resulting in incisor germ splitting (*Häärä et al., 2012*) or in a partial rescue of the lower R2 rudimentary bud (*Sadier et al., 2019*). Culturing upper molars, we have occasionally observed such a rescue of the upper R2 bud which starts to form an independent tooth cap (*Figure 4M*). This occurred much more frequently in DUHi than in FVB mice (*Figure 4K–M*; Fisher exact test; p=0.015). This is consistent with R2 being partially rescued in the DUHi mice.

We concluded that the activation-inhibition balance differs between the two strains. Next, we asked if and how these differences could explain an anterior elongation of the adult M1 that is specific to the upper molar.

## Lower and upper molar developmental systems consistently differ in their dynamics, regardless the strain

In a previous study, we had put forward differences between the lower and upper molar developmental system in CD1 mice (*Sadier et al., 2019*). In both strains, we found similar lower-upper jaw differences as seen in CD1 mice, namely 1) R2 persisted longer in the upper than in the lower jaw, as tested via a Fisher exact test on samples for which data were available for both upper and lower jaws of the same embryo (p=0.04). 2) The M1 *Shh* expression zone increased in size in upper as in lower jaw, but in the lower jaw only it encompassed the zone of the R2 signaling center (compare *Figure 2Q–T* and *Figure 2—figure supplement 1Q–T*). Thus, the spatio-temporal dynamics and fate of the R2 signaling center relative to M1 signaling center differs between the two jaws, regardless of difference in activation-inhibition mechanisms between strains. This can be considered a conserved developmental property of the lower and upper developmental systems. This may be the foundation for the lower and upper jaw developmental system reacting non-linearly to a same genetic change between FVB and DUHi mice: the increase in R2 signaling center persistence may be all the stronger as the R2 signaling center is already more persistent in the upper jaw and preserved from an early fusion with the M1 signaling center. But why would this result in anterior elongation in the upper molar only? Answering this question requires that we make comparisons between the two jaws in order to reveal how the R2 bud may contribute to the first molar.

## The premolar signaling center region contributes to different parts of the lower and upper first molar

In order to compare developmental relationship between the R2 bud and M1 in the upper and lower jaw of the DUHI and FVB strain, we genetically tracked the fate of R2 signaling center in the developing molar. Using a tamoxifen-inducible Cre/LacZ line, we were able to induce the marking of *Shh*-expressing cells and their descendants at different timepoints during tooth development. In practice, cells expressing *Shh* at the given timepoint of tamoxifen injection (in practice, a time window corresponding to tamoxifen elimination that may be 24–48 hr) recombine a lacZ transgene, so that these cells and their descendants will then be positive (blue) to a X-gal staining. By inducing the tamoxifen at different timepoints (*Figure 5A,B*, *Supplementary file 3* for a summary of all experiments), we could mark cells that descend from either the R2 signaling center and the M1 signaling center population (*Figure 5C,F*, treatment at 12.5 dpc) or from the M1 signaling center population only (*Figure 5D–H*, treatments at 13.5 and 14.5 dpc). This was done in a CD1 background, where the dynamics of R2 and M1 signaling centers is well known (*Lochovska et al., 2015*; *Prochazka et al., 2010*; *Sadier et al., 2019*) and molars are morphologically intermediate between FVB and DUHi mice (*Figure 1*). For upper molars treated during R2 signaling center activity, the anterior part of the tooth is marked at 17.5 dpc (*Figure 5C*). However, this anterior region is unmarked when tamoxifen is applied later, during M1 signaling center activity only (*Figure 5D–E*). Thus, the fate of the cell populations of R2 signaling center is the anterior region of the first upper molar. In the case of the first lower molar, the same region is stained in all three conditions (*Figure 5F–H*), consistent with R2 signaling center being overwritten by the mature M1 signaling center (*Sadier et al., 2019*). Moreover, the anterior part of the first molar is unstained. Thus, in the lower molar, the R2 signaling center does not contribute specifically to the most anterior part. From these results, we can deduce that a change in R2 size and R2-M1 centers distance, as seen in DUHi upper jaw, will directly elongate the anterior part of the upper M1 (*Figure 5C*). In contrast, the more modest change seen in the DUHi lower jaw may elongate the M1, but not specifically from its anterior part (compare *Figure 5I* with *Figure 5D–E*). We conclude that intrinsic differences between lower and upper M1 developmental systems are responsible for their different reactions: marked anterior elongation in the upper molar and a more discrete and isometric elongation of the lower molar (see *Figure 5I*).

## Developmental variation is higher in DUHi than FVB and peaks at the stage of R2-M1 coexistence

A brief examination revealed that DUHi samples display different states for similar embryonic age (*Figure 3* and S7). This suggested that the two strains may exhibit different degree of developmental variability, which we aimed to quantify. This requires disentangling differences in developmental state due to error in embryonic age estimation (e.g. two embryos are different because they were

erroneously attributed the same age) from real differences due to developmental variation (e.g. two embryos, effectively of the same age, display different states). We developed a method to measure developmental variation, which is described in detail in the Materials and methods section. This method utilises the developmental state scoring system discussed previously. The method works by identifying cases where samples differ greatly in terms of developmental state despite being approximately equally old and allows for statistical comparison of the degree of developmental variation present, *via* a Wilcoxon rank-sum test. We applied this method to determine whether there is greater developmental variation present in DUHi upper molars than FVB upper molars and in DUHi lower molars than FVB lower molars (*Figure 6A* and *Figure 6—figure supplement 1A*). Our method yielded a highly significant result in both cases, ($p<0.001$ for Wilcoxon tests) where DUHi was more variable than FVB in both cases. Therefore, not only do the developmental trajectories of these two strains differ from one another, but the degree of variation within each strain is not equivalent.

Having established that DUHi has more overall developmental variation than FVB, the next step was to develop a method to examine the temporal profile of developmental variation. This was achieved by tracking the difference in developmental state between each embryo and all other embryos of the same strain that were close in age (less than 0.25 days difference in computed embryo age). The local regression line through the developmental state difference present at a given time was then plotted, and is seen in *Figure 6B* (upper molars) and *Figure 6—figure supplement 1B* (lower molars). A similar pattern is seen in both upper and lower molars, whereby there is a small peak in developmental variation between computed age 14–15 $_c$dpc/weight: 150–250 mg, followed by a decrease in developmental variation. This peak is considerably greater, in both duration and magnitude, in DUHi than FVB samples. Some differences exist between the upper and lower molars, however. In the upper molars, it corresponds to the period when two signaling centers coexist and the M1 signaling center expands (14.5–14.75 $_c$dpc/Weight: 200–220 mg, *Figure 6B*). In the lower molar, it also matches this period in DUHi mice (around 14.5 $_c$pdc/Weight: 200 mg, *Figure 6—figure supplement 1B*), but rather corresponds to an earlier variability in termination of R2 signaling for FVB mice (around 14.1 $_c$pdc/Weight: 160 mg).

## The DUHi strain with greater developmental variation has more variable adult molars

Having established that DUHi embryos display greater variation during development than is seen in their counterparts at equivalent stages in FVB, the next step was to examine whether this developmental variation would be reflected by greater variation in adult morphology. The variation in length and width in DUHi, FVB and CD1 adults was examined for both upper and lower first molars and is shown in *Figure 6C* and *Figure 6—figure supplement 1C*. Although they are also an inbred strain, DUHi mice show greater variation in upper molar length than FVB ($p=0.095$), to an extent comparable to the outbred CD1 mice. The variation in DUHi upper molar length is consistent with the large degree of variation in R2/M1 development in DUHi embryonic upper jaws. Consistent with the developmental data, this variation in length is specific to the upper molars: In the lower molars, DUHi individuals show little variation in length, albeit large variation in width (*Figure 6—figure supplement 1C*).

## Discussion

Evolutionary trajectories are not random, but tend to take preferred routes. Several factors can explain this, among them the ability of developmental systems to vary in particular directions. Then variation typically follows similar trends at different levels, from inter-individual differences to population differences, up to species differences. This underpins the concepts of evolutionary 'lines of least resistance' (e.g. *Schluter, 1996*) or 'evolutionary predictability' inferred from developmental systems rules (*Kavanagh et al., 2007*; *Salazar-Ciudad and Jernvall, 2010*). Lines of least resistance were primarily thought of as arising from genetic constraints *Schluter, 1996*), whereas the second concept more directly refers to variational properties of developmental systems. Both relate to the old concept of 'developmental constraints' (e.g. *Smith et al., 1985*). These concepts are typically tested by matching the variation at different levels (e.g. population/macroevolution) or by simulating it with developmentally realistic models. However, they have rarely been tested by directly examining the variation in developmental systems and its systemic basis, at least for mammalian models.

Here, we used a comparative approach, focusing on fine-scale developmental differences that show evolutionary-relevant variation to unravel the generative principles underlying shape variation.

We have used inter-strain variation as a proxy for natural variation. We think this choice is relevant for three reasons: 1) We show that the morphological variation between FVB and DUHi mice recapitulates the morphological variation in the *Mus genus and beyond* 2) The DUHi mice are a product of artificial selection in the lab for large body size, starting from six different mouse strains (*Bünger and Herrendörfer, 1994*; *Bünger et al., 1982*). It thus also recapitulates the correlation between anterior elongation and large body size seen in natural populations of *Mus musculus domesticus* (*Renaud et al., 2011*). 3) Lab mice were derived from a mix of several mouse subspecies (*Mus musculus domesticus, Mus musculusmusculus, Mus musculus castaneus*). This original substantial amount of genetic variation being now split between mouse strains, it is no surprise that different mouse strain could recapitulate trends seen in the *Mus* genus.

We show that the anterior development of the first upper molar varies between DUHi and FVB strains. We show inter-related differences such as, in DUHi mice compared to FVB mice, the larger size of R2 rudimentary bud, the longer persistence of its signaling center, a posteriorisation of the M1 signaling center, a marked tendency for R2 to individualise in culture and differences in gene expression. A similar tendency, although less marked, is seen in the DUHi lower molar. Collectively, these results show that these two strains differ in the settings of the activation-inhibition mechanisms patterning the tooth signaling centers. Finding different settings of activation-inhibition mechanisms between two mouse strains is not surprising since mapping studies have found genetic variation segregating with relative molar proportions (*Navarro and Murat Maga, 2018*). We have indications that the balance is shifted for the BMP4 pathway, with the expression levels of BMP4 targets suggesting overall weaker BMP4 activity in DUHi mice, which would mean reduced activation in DUHi mice (activation is taken here in a broad sense, and may include different mechanisms, as suggested in the results). Although R2 partial rescue in a context of reduced activation may appear counterintuitive at first glance, it is in line with our recent study showing that reduced activation in culture or in the *Edar* mutant favors the R2 bud in its competition with the M1 signaling center and thereby tends to rescue it *Sadier et al. (2019)*. We note that the narrow morphology of DUHi (lower and upper) molars as compared with the broad, massive morphology of FVB molars is also consistent with decreased levels of activation in this strain. We have noticed in a study of Orkney house mouse populations that the presence of the anterior cusp was associated with a decrease in overall cusp complexity, again consistent with overall decreased level of activation (*Renaud et al., 2018*). In conclusion, our results thus suggest that the two lines of least resistance seen in murine rodents: the anterior elongation of the upper molar, and the variation in shape seen in both molars are developmentally coupled by a common setting of the activation/inhibition balance. Further work will be necessary to disentangle the different mechanisms that contribute to this balance for example, as proposed in *Sadier et al. (2019)* and reveal the basis for the difference between DUHi *and* FVB.

Our study also sheds light on the developmental reasons why these variations in activation-inhibition mechanisms would turn into elongation of the anterior part of the adult molar, specifically and repeatedly in the upper first molar. First, we show that the R2 signaling center is intrinsically stronger in the upper jaw, remains independent of the M1 signaling center and contributes to the anterior part (cervical loop) of upper M1. This is contrasted with the lower jaw, where R2 bud is smaller, included in the mature M1 signaling center (so-called pEK) and R2 signaling center cells do not contribute to the anterior cervical loop, but to a more central part of the molar. As a consequence, subtle variations in activation-inhibition mechanisms that would affect R2 signaling center could specifically impact the anterior part of the upper M1 only, whereas this effect could be either buffered in the lower M1 or spread on the whole tooth (DUHi lower M1 are longer than FVB). Secondly, upper molar development appears to be inherently more variable than lower molar development. DUHi and FVB mice are more different in terms of upper molar development than they are in terms of lower molar development; variation in R2 signaling center persistence, in anterior region size and M1 positioning are all stronger for upper than lower molars. This intrinsic variability of the upper molar is also apparent in the phenotypic variability of the DUHi mice. Although these mice are inbred, the upper molar is much more variable in length and this is correlated, again, with greater developmental variability in upper R2 signaling center (we note, however, that the lower M1 of DUHi mice is highly variable in width, *Figure 6—figure supplement 1*: this might indicate that the lower M1 also reacts to changes in R2, albeit very differently). This suggests that upper molar

development is intrinsically unstable, especially when the tuning of activation-inhibition parameters comes in the 'DUHi range'. In summary, we provide evidence for developmental particularities of the first upper molar, explaining why it responds to variations in activation-inhibition parameters by varying its anterior morphology. We thus uncovered the basis for an evolutionary line of evolutionary least resistance in murine rodents, leading to repeated morphological evolution of the first upper molar in mice.

A common conception is that developmental variation is minimal at early stages and increases over developmental time. Thus, morphological changes are assumed to result from small changes in development, which result from yet smaller changes in earlier development. This 'inverted funnel' bears a certain resemblance to Von Baer's law of embryonic divergence, and to the 'hourglass model' of interspecific developmental similarity (*Abzhanov, 2013*; *Irie and Kuratani, 2014*). Under this view, mild variations in adult phenotypes should result from almost undetectable or at least late-detectable variation in development. Here, our work identifies strong variation in early tooth development between strains (morphology of the tooth germ and dynamics of signaling centers) in tandem with relatively mild variation in adult phenotype. Besides influencing our view of developmental variation, this has implications for developmental biology practices, especially in the mouse model in which access to embryo is limited for both ethical and cost reasons. For example, heterozygotes are often used as controls for developmental genetic studies, where heterozygotes do not display an obvious phenotype in adults, because it is assumed that the development underlying that phenotype proceeds normally. However, our work here finds greater variation in early development than in the adult phenotype. In systems like that observed here, heterozygotes may have important differences in early development, making their use as controls unreliable. We also note that discrepancies between observations made in different labs will a priori not be attributed to a difference in the wild type strain used in each lab. In this context, it is interesting that the recognition of R2 vestigial bud presence in mouse molar development was a matter of debate for several years, and R2 is much less transient and discrete in the CD1 strain, where R2 was first identified, than in FVB, in which much heavier sampling is needed to catch the short developmental window when it is present. Therefore, it is well possible that differences between strains worked to obscure the debate. In conclusion, we believe that enhancing the focus on developmental variation will be important to move on from overly simplistic views of developmental variation that more or less consciously influence our practices in biology.

## Materials and methods

**Key resources table**

| Reagent type (species) or resource | Designation | Source or reference | Identifiers | Additional information |
|---|---|---|---|---|
| Strain, strain background (*Mus musculus*) | FVB | Charles River, France | | inbred |
| Strain, strain background (*Mus musculus*) | CD1 | Charles River, France | | outbred |
| Strain, strain background (*Mus musculus*) | DUHi | MRC Mary Lyon centre, Oxfordshire, UK | | inbred |
| Strain, strain background (*Mus musculus*) | B6.129S6-Shh < tm2(cre/ERT2)Cjt>/J | Jackson Laboratory, Maine, USA | | Originally in C57Bl6 background. Backcrossed to CD1 |
| Strain, strain background (*Mus musculus*) | B6.129S4-Gt(ROSA)26Sortm1LacZSor/J | Jackson Laboratory, Maine, USA | | Originally in C57Bl6 background. Backcrossed to CD1 |
| Commercial assay or kit | TruSeq RNA Sample Prep Kit v2 | Illumina | RS-122–2001 | Non stranded protocol |

*Continued on next page*

*Continued*

| Reagent type (species) or resource | Designation | Source or reference | Identifiers | Additional information |
|---|---|---|---|---|
| Commercial assay or kit | DISPASE II (NEUTRAL PROTEASE, GRADE II) | ROCHE/SIGMA | 4942078001 | Used at 10 mg/ml in Hepes/KOH 50mM ph7.7 ; NaCl 150 mM. Incubation at 37°C from 45 min to 1h15 depending on stage. |
| Antibody | ANTI-DIGOXIGENIN AP-CONJUGATE, from sheep | ROCHE/SIGMA | 11093274910 | Fab fragments from polyclonal anti-digoxigenin antibodies (from sheep), conjugated to alkaline phosphatase. Used 1/1200ie |
| Commercial assay or kit | BM PURPLE AP SUBSTRATE, PRECIPITATING | ROCHE/SIGMA | 11442074001 | |
| Commercial assay or kit | DIG RNA Labeling Mixture, 10x | ROCHE/SIGMA | 11277073910 | |

## Animal husbandry and ethical commitment

DUHi mice were raised at the PBES; cryopreserved embryos had been obtained from MRC Mary Lyon centre, Oxfordshire, UK. FVB and CD1 mice were purchased from Charles River company.

C57BL/6 mice carrying tamoxifen-inducible Cre fused with the *Shh* allele (B6.129S6-Shh < tm2 (cre/ERT2)Cjt>/J) and Cre recombinase-sensitive transgenic mice cB6.129S4-Gt(ROSA)26Sortm1-LacZSor/J) containing *LacZ* (beta-galactosidase) inserted into the *Gt(ROSA)26Sor* locus were used for the cell fate tracing study. The breeding pairs were purchased from the Jackson Laboratory (Maine, USA). The mice were genotyped using the Jackson Laboratory's protocols.

This study was performed in a strict accordance with European guidelines (2010/63/UE). It was approved by the CECCAPP Animal Experimentation Ethics Committee (Lyon, France; reference ENS_2014_022), by the Professional committee for guarantee of good life-conditions of experimental animals at the Institute of Experimental Medicine IEM CAS, Prague, Czech Republic) and by the Expert Committee at the Czech Academy of Sciences (permit number: 027/2011).

## Morphometric analyses

First upper and lower molars of a set of adult mice from the strains DUHi (19), FVB (11) and CD1 (17) were pictured using a Leica MZ 9.5 stereomicroscope. These teeth were compared with the variation observed within the murine rodents (Murinae). Their morphological diversity was documented by a set of specimens from the Museum National d'Histoire Naturelle (Paris, France) covering the main divisions of the group (*Lecompte et al., 2008* ): Rattini with *Rattus whiteheadi* (5), *Micromys minutus* (9), *Berylmys* sp. (4), *Leopoldamys sabanus* (10), *Bandicota indica* and *bengalensis* (11) and *Sundamys muelleri* (9); Arvicanthini with *Golunda ellioti* (6), *Lemniscomys barbarus* (6), *Oenomys hypoxanthus* (5), *Rhabdomys pumilio* (5), *Arvicanthis niloticus* (5) and *Aethomys chrysophilus* and *namaquensis* (10); Praomyini with *Mastomys chrysophilus* (6) and *Praomys tullbergi* (7); Murini with *Mus cervicolor* (5), *Nannomys setulosus* (6), and *Nannomys mattheyi* (4). Apodemini were represented by *Apodemus sylvaticus* (13; data from *Renaud et al., 2009*). The sampling was completed with wild house mouse (*Mus musculus domesticus*) samples, documenting the continental and insular variation: Gardouch, France (68), Montpellier, France (13); Eday, Orkney, United Kingdom (18), Fango, Corsica, France (7), Vaitella, Corsica, France (24) and the islet Piana off Corsica, France (7) (data from *Ledevin et al., 2016*; *Renaud et al., 2011*). Maximal length and width were automatically extracted, together with 64 points along the outline, using the image analyzing software Optimas 6.5.

The morphological variance in the total sample, including laboratory strains and wild species of murine rodents (200 specimens, with 198 first upper molars (UM1) and 192 first lower molars (LM1)) was summarized using principal component analyses (PCA) (one for the upper and one for the lower

molars) on the variance-covariance matrix of the shape coefficients delivered by an outline analysis of the 2D occlusal surface (see *Renaud et al., 2009*; *Renaud et al., 2011*). Fourteen variables were considered for both the first upper molar (UM1) and the first lower molar (LM1); all were standardized by the size of the respective tooth and corresponded to shape only. Maximum length and width of the outline were also measured and allowed the estimation of the overall elongation of the tooth (Length/Width ratio).

### Embryo harvesting and staging

Mouse females were mated overnight and the morning detection of a vaginal plug was taken as proof of coitus, noon being taken 0.5 days post coïtum (dpc). We used a different day/night regime 12 hr apart to obtain embryos every half day. Pregnant females were sacrificed via cervical dislocation and embryos were harvested on ice and weighted.

### Dental epithelial dissociation

Embryos were dissected in Hank's medium to separate upper from lower molars and then treated in Dispase II (Roche) 10 mg/mL at 37°C for 1 to 2 hr, depending on embryonic stage. Dental epithelium was then carefully removed and fixed overnight in paraformaldehyde (PFA) 4%.

### In situ hybridization

*Shh* probes were transcribed from a plasmid described by *Echelard et al. (1993)*, by means of in vitro transcription with the incorporation of digoxigenin-ddUTP, using a premixed DIG RNA labelling mix (Roche). In situ hybridisation was performed with a conventional protocol. The antibody utilised was an anti-DIG antibody coupled with alkaline phosphatase (Roche); the chromogenic substrate used was BM Purple ready-to-use NBT/BCIP (Roche). Samples were examined with Leica M205 stereomicroscope. Images were taken with the Leica Application Suite 4.1 software package.

### Modeling embryonic age

In this study, we needed a numeric estimation of the embryonic age of embryos (i.e. an age reflecting the progress in embryogenesis). In a given strain, harvesting age estimated from the calculation of days-post-coitum provides a very rough estimation, because of important differences between litters (standard range ±0 to 0.5 days variation in embryonic age, notably due to difference in fertilization and implantation time) and because it does not take into account the slight variation in developmental stage within litters (standard range from 0 to a quarter day). A combined 'age/weight' staging has been recommended previously (*Peterka et al., 2002*). Embryonic body weight provides a much better numeric estimation than age in dpc alone (*Pantalacci et al., 2009*; *Peterka et al., 2002*). The embryonic body weight alone is especially reliable within litters (at the stages examined here) but less reliable between litters, presumably because the nutritional status differs from one pregnancy to the other, causing embryos of similar embryonic age to have smaller or larger body weight. Age in dpc can help to correct for this, since embryos with similar body weight and similar harvesting age will have higher chance to have reached similar developmental stage (similar embryonic age), while embryos with similar body weight but different harvesting age in dpc will have higher chances to be at different embryonic age. On top of these intra-strain differences, there are differences between strains: embryos sampled at a given number of days post-coitum were not of the exact same embryonic age range in each strain. In order to provide a measure of embryonic age, we built a model that could estimate the embryonic age from the body weight and dpc, taking into account intra and inter litter variations and applying a correction effect for the latter. This model and its construction are described in detail in Appendix 1. The code is available at: https://github.com/msemon/cdpc (*Sémon, 2020a*; copy archived at https://github.com/elifesciences-publications/cdpc).

### Sample classification and scoring

The developmental state of all samples was assessed by combining four separate developmental criteria. Two are related with Shh expression: 1) *Shh* expression in the R2 (rudimentary premolar) zone, 2) *Shh* expression in the M1 (first molar) zone, and two are purely morphological criteria: 3) the bud-cap transition and 4) the appearance of a protrusion, visible in the dissociated epithelium at the site

of the R2 signaling center, but only in the latest stages of Shh signaling and after cessation of Shh signaling. For each criterion, the samples were scored as one of 2 or 3 states. This scoring system is summarized in *Supplementary file 1*. Two tables with all analyzed embryos, with their weight, age in cdpc, information on litter and scored characteristics, and the code used to analyze these data is provided on github at: https://github.com/luke-hayden/tree/master/dvpap/devstate (*Hayden, 2020*; copy archived at https://github.com/elifesciences-publications/dvpap).

### 3D reconstruction

Dissected tooth germs were fixed overnight in 4% PFA and dehydrated through a methanol series. *In toto* immunolocalization protocol was adapted from *Ahnfelt-Rønne et al. (2007)*. Following incubation in methanol added with H202 5% and DMSO 10% for 4 hr at room temperature, they were rehydrated, blocked with serum and incubated successively with an anti-laminin5a antibody (overnight,1/800, kind gift from Jeff Miner, *Miner et al., 1997*) and a Dylight 549 conjugated Donkey Anti-rabbit antibody (overnight 1/200, Jackson immunoresearch). Following dehydration, they were clarified and mounted in BABB as described in *Yokomizo and Dzierzak (2010)*. They were imaged with a Zeiss LSM710 confocal microscope at the PLATIM (Lyon, France). The basal membrane labeled by the antibody was delineated semi-manually and reconstructed with the AMIRA software.

### First molar germs RNA sequencing

RNA-Seq were made on 12 carefully dissected embryonic lower and upper first molar germs, from DUHi (embryo weight: 196, 219 and 239 mg) and FVB (195, 215 and 233 mg) strains. Following dissection in culture medium, tooth germs were stored in RNA later at −20℃. RNA was extracted with RNAeasy micro kit (Qiagen), and controlled with Q-bit (Invitrogen) and Tapestation (Agilent technologies). RNA-Seq samples were prepared following the TruSeq RNA Sample Preparation v2 Guide, starting from 100 ng of total RNA of top quality (RINe > 9.5). Sequencing was performed with the Illumina HiSeq 4000 system (single-end 50 bp reads).

### Detection of differentially expressed genes with DESeq2 package

Reads were then mapped to the mouse genome using Kallisto (*Bray et al., 2016*, version 0.44.0, options -l 200, -s 20). The reference cDNA sequences and annotation files for *M. musculus* are based on C57B6 strain. They were collected from Ensembl 88 (10 5129 cDNAs, [*Zerbino et al., 2018*], GRCm38). Reads were independently mapped to the FVB/NJ strain cDNAs, collected from Ensembl strains 94, using biomart (10 1520 cDNAs, strain FVB_NJ_v1, accession GCA_001624535.1). Tximport was used to import and summarize transcript-level estimates at gene level (version 1.6, *Soneson et al., 2016*). Differentially expressed genes were detected with DESeq2 (*Love et al., 2014*), version 1.18.1) with classical one-factor design, and using FDR significance threshold = 0.05. 19202 genes are in common between FVB and reference strain C57B6 and have a MGI annotation. Out of these genes, 2234 genes (11.6%) presented a significant difference of expression between the mapping on the reference strain C57B6 and the mapping on FVB strain (DESeq2, adjusted p-value<0.05, considering the mapping effect that is with 12 replicates). This is presumably a mapping artifact, due to the sequence divergence between mouse strains. These genes were removed, and the remaining 16,968 genes were retained for further analysis. 3619 genes were found to be differentially expressed between the two strains taking into account the jaw of origin (lower/upper) (~jaw + Strain). Processed data with statistics are provided in *Supplementary file 2*. Raw and processed data were deposited in NCBI Gene Expression Omnibus (GEO, accession number GSE135432; https://www.ncbi.nlm.nih.gov/geo/query/acc.cgi?acc=GSE135432). The mapping data (by Kallisto, on each reference strain as discussed in the text), R source code and parameters are available on github at: https://github.com/msemon/trDUHi_FVB (*Sémon, 2020b*; copy archived at https://github.com/elifesciences-publications/trDUHi_FVB).

### Comparison of BMP4 and Wnt pathway signaling activity in DUHi and FVB mice based on transcriptomic data

We used a list from supplementary data published by *O'Connell et al. (2012)* describing regulatory interactions for BMP4 and Wnt pathways in tooth epithelium and tooth mesenchyme at different developmental stages, by combining data mining with results of their own perturbation experiments.

For BMP4 pathway, it describes up or downregulation (+: upregulated, -: downregulated, o: no change) of presumptive target genes upon BMP4 treatment (perturbation +) or BMP4 knockout (perturbation -). For Wnt pathway, it describes up or downregulation of presumptive target genes upon inhibition of Gsk3b (i.e. Wnt pathway activation), *Ctnnb1* overexpression (i.e. Wnt pathway activation), *Ctnnb1* knock-out (i.e. Wnt pathway inhibition), *Dkk1* overexpression (i.e. Wnt pathway inhibition), *Lef1* knock-out (i.e. Wnt pathway inhibition), or treatment with different Wnts. First, we checked if genes of this list were differentially expressed in the above analysis. For those that were DE, we compiled the O'Connell table to determine if the gene was a positive, a negative or not a target of the pathway. A gene that is upregulated upon pathway activation, or downregulated upon pathway inhibition was considered a positive target. A gene that is downregulated upon pathway activation, or upregulated upon pathway inhibition was considered a negative target. When data were conflicting between tissues (e.g. positive target in epithelium, negative target in mesenchyme; 7 genes, e.g. *Dlx1*), the gene was excluded from the analysis, because in our analysis the whole tooth germ is examined. When data were conflicting between different sources (two genes: *Egr1*, *Ptch1*), we kept the result obtained by *O'Connell et al. (2012)* because 1) these are transcriptomic data and 2) most interactions described in the table are from this study only or are confirmed in this study, and thus the result obtained for this gene has more chance to be consistent with results for other genes in the table. The resulting table is shown in *Supplementary file 2*.

## Fate mapping of *Shh* expressing cells using X-gal staining

The strain B6.129S6-Shh < tm2(cre/ERT2)Cjt>/J was reciprocally crossed with a reporter strain containing LacZ inserted into the *Gt(ROSA)26Sor* locus in order to mark the cell population expressing *Shh* from the time of the tamoxifen injection into pregnant female mice. Pregnant female mice were injected intra-peritoneally with tamoxifen at E12.5 (when *Shh* is expressed in the R2 expression domain and early M1 expression is not yet apparent), E13.5 (when Shh expressing domain in R2 finishes its activity and early M1 signaling center starts to be apparent posteriorly) or at E14.5 (when only M1 signaling center express Shh). Tamoxifen was administrated in a dose of 0.225 mg/g of body-weight (*Hayashi and McMahon, 2002*). Such a concentration is not hazardous for pregnant mice or embryos and is sufficient for the fast activation of recombination. The embryos were harvested at 17.5 dpc, 72, 96 or 120 hr after tamoxifen application and beta-galactosidase activity was detected on whole embryos or dissociated epithelia of upper and lower cheek region. The X-gal (Sigma) concentration in the staining buffer was 3 mM. Samples with positive staining were post-fixed in PFA (4%) overnight. After post-fixation, the samples were washed in PBS and photographed using a Leica MZ6 stereomicroscope equipped with a Leica EC3 digital camera (Leica Microsystems GmbH, Wetzlar, Germany). Data are summarized in *Supplementary file 3*.

## Organotypic culture

The upper molar region of 13.0 dpc FVB or DUHi embryos were dissected and cultured according to standard methods described in *Kavanagh et al. (2007)*. Tooth culture was stopped after 40 hr and imaged using a Leica MZ6 stereomicroscope equipped with a Leica EC3 digital camera (Leica Microsystems GmbH, Wetzlar, Germany).

## Measurement of developmental variation

Measuring developmental variation is a complex task; it requires that we can measure factors that change with time as development proceeds. If we consider how a developmental system proceeds along its trajectory, we will expect that it changes gradually over time, where a given sample is most similar to those at the closest time-points. However, where developmental variation is present, we expect to find pairs of samples that are of the same embryonic age but differ markedly in form. So, for each strain and for both upper and lower molars separately, we took all of our set of samples, determined their computed embryonic age (in $_c$dpc) and their developmental state, then took all possible pairs of samples where both members of the pair were close in age (less than 0.25 $_c$dpc difference). For each pair of samples, we can then compute a pairwise developmental distance: the distance between the two samples in terms of developmental state, computed as the sum of the score difference obtained for each four developmental criteria (*Supplementary file 1*). For each strain for both upper and lower molars, we could then plot the distribution of these pairwise developmental

distances. Finally, in order to compare total developmental variation between strains, we subjected these pairwise developmental state differences (between pairs of samples at $_c$dpc) to a Wilcoxon rank-sum test. Using this method, we can measure the degree of developmental variation found in a set of samples and compare between strains. The code is provided on github at: https://github.com/luke-hayden/dvpap/tree/master/devstate (*Hayden, 2020*; copy archived at https://github.com/elifesciences-publications/dvpap).

## Measurement of developmental variation over time

We first calculated pairwise developmental state distances for each sample in relation to all nearby samples, over computed embryonic time. Then to obtain developmental variation over time, we used locally estimated scatterplot smoothing (LOESS), a non-parametric regression method to plot a complex curve through many data points, weighting the contribution of data points according to their proximity to the point of estimation. The code is provided on github at: https://github.com/luke-hayden/dvpap/tree/master/devstate (*Hayden, 2020*; copy archived at https://github.com/elifesciences-publications/dvpap).

## Other statistical analyses

The statistical significance of differences in the timing of developmental events was tested using logistic regression (embryo weight as a predictor of state, with strain as an additional preictive factor), examining changes in the scoring of a developmental criterion (four criteria scored, see previously) in relation to computed embryonic age. The statistical significance of differences in the sizes of various morphological features was tested using Student's t-test. Fisher's test was used to test differences in the relative rarity of a developmental state within a given window. The code is provided on github: https://github.com/luke-hayden/dvpap/tree/master/devstate (*Hayden, 2020*; copy archived at https://github.com/elifesciences-publications/dvpap). See also *Supplementary file 4*.

All statistical analyses were carried out using the R statistical environment (*R Development Core Team, 2014*), version 3.2.3. Packages used included ggplot2 (*Wickham, 2009*), reshape2 (*Wickham, 2007*) and phytools (*Revell, 2012*).

## Acknowledgements

We acknowledge the contribution of the Plateau de Biologie Expérimentale de la Souris (PBES), more especially Marie Teixeira, Farida Henry and Céline Angleraux for DUHi mice recovery and breeding, and the Plateau Technique Imagerie/Microscopie (PLATIM) of SFR Biosciences Gerland-Lyon Sud (UMS344/US8), more especially Claire Lionnet and Christophe Chamot for their support and help with 3D reconstructions. We are grateful to Jeff Miner for the anti-laminin5 antibody and Pascale Chevret for her expertise in the phylogeny of murine rodents. This work was supported by Agence Nationale de la Recherche (ANR-11-BSV7-008 'Bigtooth') and Fondation pour la recherche médicale (FRM; SPF20140129165). Salaries were supported by CNRS for SP and SR, Ecole Normale Supérieure de Lyon for MS.

This work was supported by the Grant Agency of the Czech Republic (14–37368G; 18–04859S) and the Czech Ministry of Education, Youth and Sports (8J19FR032) and by the Charles University (Research Program Progress Q29).

## Additional information

### Funding

| Funder | Grant reference number | Author |
|---|---|---|
| Agence Nationale de la Recherche | ANR-11-BSV7-008 | Sabrina Renaud<br>Sophie Pantalacci |
| Fondation pour la Recherche Médicale | SPF20140129165 | Luke Hayden |
| Grant Agency of the Czech Republic | 14-37368G | Renata Peterkova |

| Ministerstvo Školství, Mládeže a T?lovýchovy | 8J19FR032 | Maria Hovorakova |
| Grant Agency of the Czech Republic | 18-04859S | Maria Hovorakova |
| Centre National de la Recherche Scientifique | Salary | Sabrina Renaud Sophie Pantalacci |
| École Normale Superieure de Lyon | Salary | Marie Sémon |

The funders had no role in study design, data collection and interpretation, or the decision to submit the work for publication.

### Author contributions

Luke Hayden, Conceptualization, Investigation, Visualization, Methodology, Writing - original draft, Writing - review and editing; Katerina Lochovska, Maurine Vilcot, Investigation; Marie Sémon, Conceptualization, Investigation, Methodology, Writing - original draft; Sabrina Renaud, Conceptualization, Investigation, Writing - review and editing; Marie-Laure Delignette-Muller, Conceptualization, Investigation, Methodology; Renata Peterkova, Funding acquisition, Writing - review and editing; Maria Hovorakova, Supervision, Funding acquisition, Investigation, Project administration, Writing - review and editing; Sophie Pantalacci, Conceptualization, Supervision, Funding acquisition, Investigation, Writing - original draft, Project administration, Writing - review and editing

### Author ORCIDs

Luke Hayden (ID) https://orcid.org/0000-0002-1939-5714
Katerina Lochovska (ID) http://orcid.org/0000-0003-4142-4531
Marie Sémon (ID) https://orcid.org/0000-0003-3479-7524
Sabrina Renaud (ID) http://orcid.org/0000-0002-8730-3113
Marie-Laure Delignette-Muller (ID) https://orcid.org/0000-0001-5453-3994
Maurine Vilcot (ID) https://orcid.org/0000-0001-8765-1364
Maria Hovorakova (ID) https://orcid.org/0000-0002-6346-6671
Sophie Pantalacci (ID) https://orcid.org/0000-0002-0771-8985

### Ethics

Animal experimentation: This study was performed in a strict accordance with European guidelines (2010/63/UE). It was approved by the CECCAPP Animal Experimentation Ethics Committee (Lyon, France; reference ENS_2014_022), by the Professional committee for guarantee of good life-conditions of experimental animals at the Institute of Experimental Medicine IEM CAS, Prague, Czech Republic) and by the Expert Committee at the Czech Academy of Sciences (permit number: 027/2011).

### Decision letter and Author response

Decision letter https://doi.org/10.7554/eLife.50103.sa1
Author response https://doi.org/10.7554/eLife.50103.sa2

## Additional files

### Supplementary files

• Supplementary file 1. Table with scoring criteria used to assess embryonic dental epithelia.

• Supplementary file 2. An excel file for transcriptomic analysis: normalized basemean for all genes with statistical support in our DE analysis, list of DE genes for padj < 0.05, list of BMP4 pathway target genes extracted from *O'Connell et al. (2012)* and their classification as activated/repressed target of BMP4 pathway, list of BMP4 target genes DE in DUHi/FVB, list of Wnt pathway target genes extracted from *O'Connell et al. (2012)*, and their classification as activated/repressed targets; list of Wnt target genes DE in DUHi/FVB.

- Supplementary file 3. Table summarizing lineage tracing experiments.
- Supplementary file 4. Tables for key statistical tests performed for this study.
- Transparent reporting form

### Data availability

Sequencing data have been deposited in GEO under accession codes GSE135432. - All data generated or analyzed during this study are included in the manuscript and supporting files. Sources and codes are available on GitHub https://github.com/msemon/cdpc; https://github.com/luke-hayden/dvpap/tree/master/devstate; https://github.com/luke-hayden/dvpap/tree/master/devmorph; https://github.com/msemon/trDUHi_FVB.

The following dataset was generated:

| Author(s) | Year | Dataset title | Dataset URL | Database and Identifier |
|---|---|---|---|---|
| Pantalacci S, Sémon M, Hayden L, Vilcot M | 2019 | Comparative study of gene expression in lower and upper first molar cap stage tooth germs of DUHi and FVB mice | https://www.ncbi.nlm.nih.gov/geo/query/acc.cgi?acc=GSE135432 | NCBI Gene Expression Omnibus, GSE135432 |

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

## Appendix 1

### Bayesian method to estimate embryonic age for mouse embryos

See also: https://github.com/msemon/cdpc (*Sémon, 2020a*).

### Background

Embryonic age in mouse embryos can be crudely estimated using the number of days post coïtum (dpc, where noon of the day when a plug is detected is equal to 0.5 days). For an equivalent age in dpc, mouse embryos show considerable inter and intra-litter variability in embryonic age, despite the absence of genetic variation in inbred mouse strains. Reasons for this are environmental effects on the embryo that are either common to the whole litter (female physiology influencing the timing of ovulation or implantation, number of embryos implanted influencing the demand on the maternal physiology) or specific to the embryo (e.g. position in the uterus determining placental efficiency).

To study fine-scale dynamics of tooth development, we needed to estimate a numeric embryonic age. Embryo weight is well correlated with embryonic age (including tooth developmental stage, *Peterka et al., 2002*) and can be used as a proxy. However, variability in embryo weight is not solely due to variability in embryonic age. Although very limited nutritional availability can slow down developmental rate, subtler limitations may only result in low body weight embryos, as compared to others with the same embryonic age. Reciprocally, exceptionally good nutritional conditions should not speed up developmental rate, but rather result in high body weight of embryos, as compared to others with the same developmental stage. In late stages, sex influences body weight (males are heavier). In our practice, we noticed that embryos and/or litters that deviate from the expected body weight for their age in dpc have more chances to be mis-estimated from their body weight only. This is the case for embryos displaying extreme body weight within a litter distribution at a given age in dpc. This is also the case for litters for which body weight poorly matches with dpc.

Therefore, we wish to build a model that could estimate the embryonic age from the body weight and dpc, taking into account intra and inter litter variations.

### Construction of the model

We built a model to improve embryonic age estimation based on observed values of embryo weight, embryo litter, and age in dpc (for the litter).

First, we consider a basic model (a deterministic one) where the embryonic age of embryos would be equal to their age in dpc. We assume that there is a log-linear relationship between weight and dpc specific to each strain (FVB or DUHi). The slope and offset of this relationship are estimated from the data for each strain separately and termed respectively 'b' and 'a' in the model. The body weight in litter '*i*' can then be modeled as in *Equation 1*

$$\text{weight.in.log}_i = a + b\text{*age.in.dpc}_i \tag{1}$$

In practice, age.in.dpc is not a precise measure of embryonic age. Different litters at the same age in dpc may be more or less developmentally advanced (inter-litter developmental effect) *Equation 2*. This induces a variability on weight.in.log in model *Equation 1* which is described in the model *Equation 2* by the addition of a stochastic term, eps.litter.dev, following a Gaussian distribution centered on 0 with a standard deviation, sd.litter.dev, that will be estimated from data.

$$\text{weight.in.log}_i = a + b * \text{age.in.dpc}_i + \text{eps.litter.dev}_i \tag{2}$$

However, different litters at the same age in dpc, and at the same embryonic age, may have different mean weights because each pregnancy will provide a specific environment to

the embryos. The addition of another stochastic term in the model, eps.litter.preg, is needed to account for this pregnancy effect *Equation 3*. We assume that this effect follows a Gaussian distribution centered on 0 with a known standard deviation (sd.litter.preg). From our practice we consider two values for sd.litter.preg: 0.05 and 0.1. The first one (0.05) corresponds to a 95% fluctuation interval of eps.litter.preg of [−0.1; 0.1] which gives in the ratio of weight [−10%; 10%], so to a realistic maximum effect on weight of 20 mg for a 200 mg embryo. The second value (0.1) corresponds to a 95% fluctuation interval of eps.litter.preg of [−0.2; 0.2] which corresponds in the ratio in weight of [−18%; 22%] so to an excessive maximum effect on weight of 40 mg for a 200 mg embryo.

$$\text{weight.in.log}_i = a + b^*\text{age.in.dpc}_i + \text{eps.litter.dev}_i + \text{eps.litter.preg}_i \tag{3}$$

The previous complexifications of the model are drawn at the level of the litter, presuming that all embryos of a given litter were at the same embryonic age. However, we know that 1) all embryos are not at the same embryonic age and 2) within a litter, weight is a very good indicator of relative embryonic age. To take advantage of this, we modeled that within a litter, embryonic age follows a Gaussian distribution, centered on the mean stage of the litter. So to describe the weight of the embryo *j* in the litter *i*, we add to the model *Equation 4* a last term, eps.embryo.dev, following a Gaussian distribution centered on 0 with a standard deviation of sd.embryo.dev estimated from the data and characterizing intra-litter variability on body weight.

$$\text{Weight.in.log}_{ij} = a + b^*\text{age.in.dpc}_i + \text{eps.litter.dev}_i + \text{eps.litter.preg}_i + \text{eps.embryo.dev}_{ij} \tag{4}$$

## Estimation of embryonic age

Once the model fitted, the development age was estimated from model 4 just by removing the pregnancy effect eps.litter.pregi and inverting the relation:

$$\text{age.dev}_{ij} = (\text{weight.in.log}_{ij} - \text{eps.litter.preg}_i - a)/b \tag{5}$$

Note that this estimation of the embryonic age (age.devij) from the body weight in log of each embryo (weight.in.logij) requires the knowledge of parameters a and b and of random effects due to pregnancy for each litter (eps.litter.pregi). Those were previously estimated from data as explained below.

## Estimation of the parameters

Parameters a, b, sd.litter.dev, sd.embryo.dev and random effects of the model were estimated from data in two scenarios for two fixed sd.litter.preg values (0.05 in the realistic scenario and 0.10 in the excessive scenario) as it is not possible to dissociate only from data the two components of inter-litter variability: variability in weight due to embryonic age and to pregnancy. Vague uniform priors were assigned to the other parameters (a, b, sd.litter.dev, sd.embryo.dev) allowing variation of each within a realistic range (see Supplementary methods *Appendix 1—table 1*)

**Appendix 1—table 1.** Medians and 95% credibility intervals for the parameters for each strain, and each scenario (termed realistic for sd.litter.preg = 0.05, and permissive for sd.litter.preg = 0.1 in the text).

**A: DUHi, permissive model**

| DUHi | sd.litter.preg=0.1 | | |
|------|------|------|------|
| | 2.5% | 50% (median) | 97.5% |
| a | 5.3937 | 5.4742 | 5.5531 |
| b | 0.4816 | 0.5402 | 0.6065 |

*Appendix 1—table 1 continued on next page*

*Appendix 1—table 1 continued*

**A: DUHi, permissive model**

| DUHi | sd.litter.preg=0.1 | | |
|---|---|---|---|
| sd.embryo.dev | 0.1556 | 0.1695 | 0.1855 |
| sd.litter.dev | 0.1376 | 0.1997 | 0.2889 |
| B: DUHi, realistic model | | | |
| DUHi | sd.litter.preg=0.05 | | |
| | 2.5% | 50% | 97.5% |
| a | 5.3928 | 5.4732 | 5.5555 |
| b | 0.4809 | 0.5391 | 0.6050 |
| sd.embryo.dev | 0.1557 | 0.1694 | 0.1854 |
| sd.litter.dev | 0.1632 | 0.2188 | 0.3003 |
| C: FVB, permissive model | | | |
| FVB | sd.litter.preg=0.01 | | |
| | 2.5% | 50% | 97.5% |
| a | 5.5019 | 5.5697 | 5.6378 |
| b | 0.4057 | 0.4622 | 0.5173 |
| sd.embryo.dev | 0.0812 | 0.0875 | 0.0946 |
| sd.litter.dev | 0.1549 | 0.2032 | 0.2683 |
| D : FVB, realistic model | | | |
| FVB | sd.litter.preg=0.01 | | |
| | 2.5% | 50% | 97.5% |
| a | 5.5019 | 5.5697 | 5.6378 |
| b | 0.4060 | 0.4611 | 0.5175 |
| sd.embryo.dev | 0.0813 | 0.0875 | 0.0944 |
| sd.litter.dev | 0.1774 | 0.2211 | 0.2817 |

Monte Carlo Markov-Chain (MCMC) techniques were used to estimate the joint posterior distribution of parameters from prior distributions and data. Computations were performed using the JAGS software via the R package rjags (*Plummer, 2016*, a runnable R script is provided in Supplementary material with data corresponding to strain FVB). Three independent MCMC chains were run in parallel. For each chain, 110,000 samples were produced. The first 10,000 were considered as burn-in phase and discarded. To avoid autocorrelation, the remaining 100,000 samples were thinned by selecting one out of 20 samples, thus keeping 5000 samples per chain. We checked the convergence again by displaying MCMC chain traces and autocorrelation plots and by computing the Gelman and Rubin's statistics as modified by *Brooks and Gelman (1998)*. For each parameter, its point estimate was defined as the median of its marginal posterior distribution, and the 95% credible interval was defined from the 2.5 and 97.5 percentiles of this distribution. The calculation of the development age for each embryo was integrated in the model was thus estimated in the same way from its posterior distribution estimated by MCMC.

## Comparison of age measurements

## Comparison of age measurements

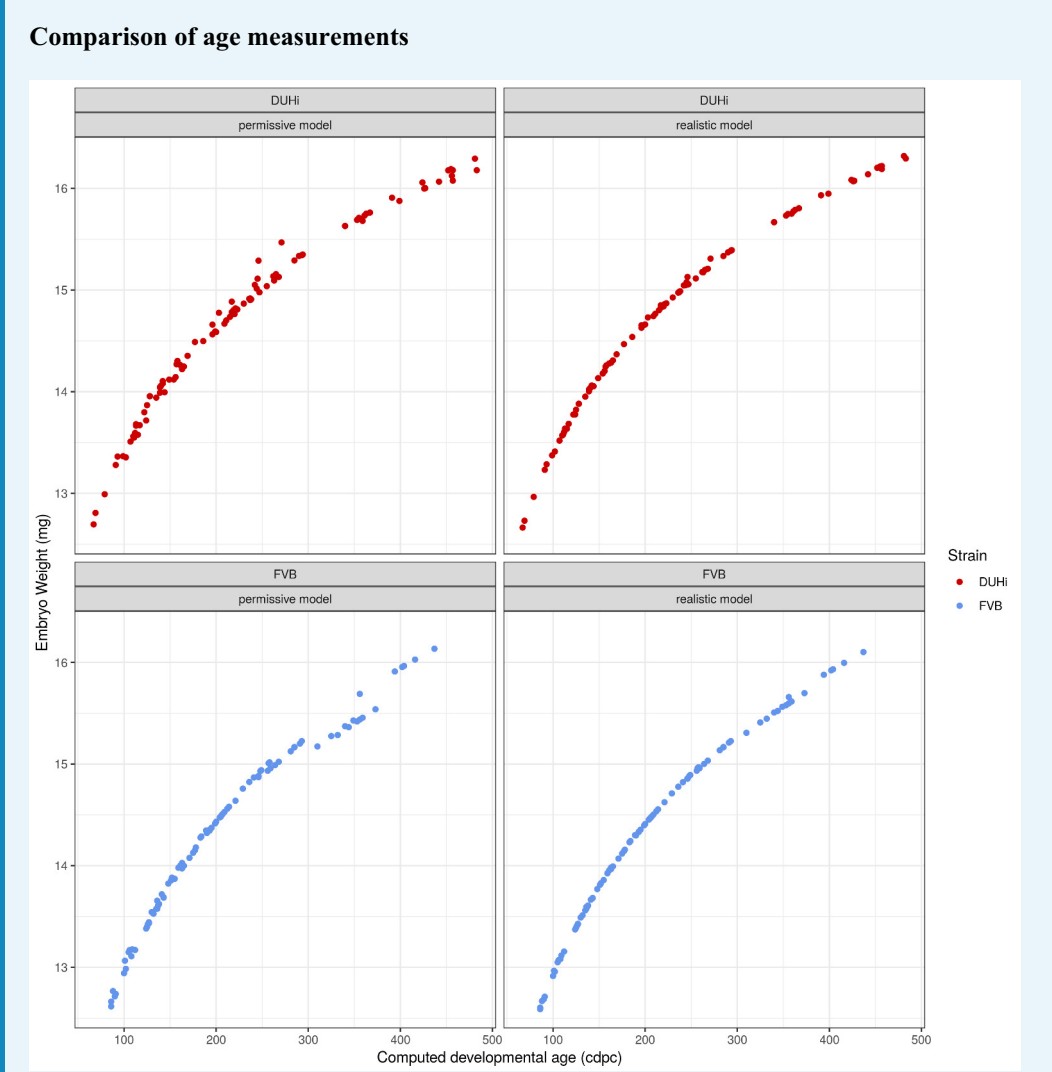

**Appendix 1—figure 1.** Comparison between two versions of computed embryonic age (realistic versus permissive), as plotted against embryo weight (mg). DUHi (red) and FVB (blue) samples are shown in separate panels.

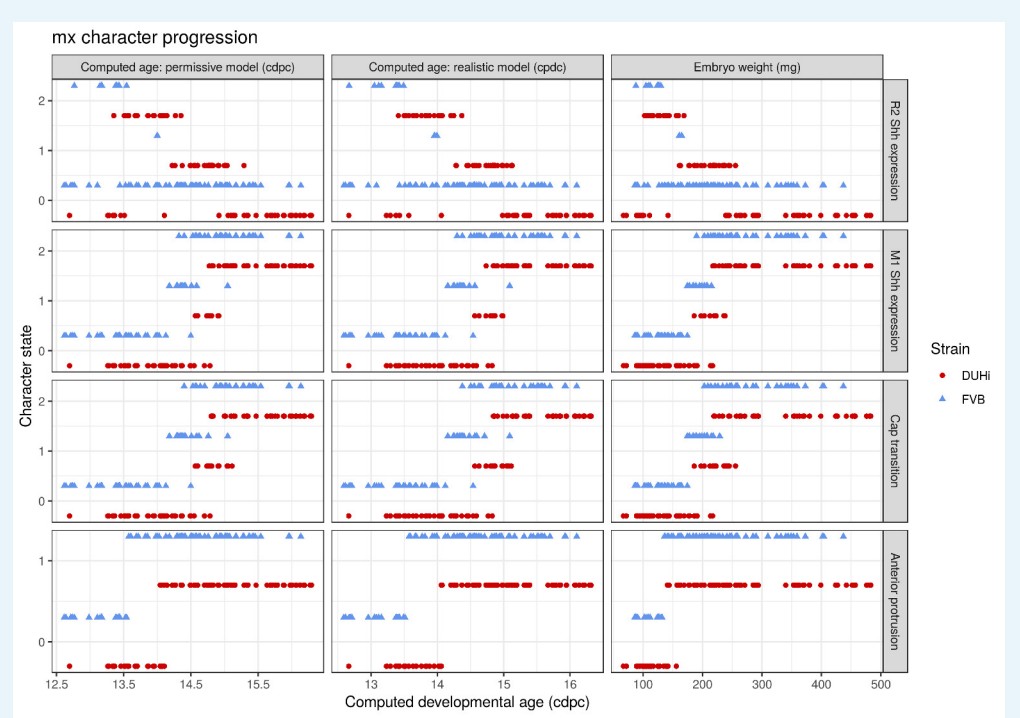

**Appendix 1—figure 2.** Comparative progression of scored characteristics in upper molar epithelia under different measures of age. Progression of four characters (R2 *Shh* expression, M1 *Shh* expression, cap transition and anterior protrusion of the dental epithelium) is depicted in samples of upper molars from two strains (FVB and DUHi). In order to allow the comparison of age measurements, the temporal axis is provided by computed embryonic age ($_c$dpc) under the realistic and permissive parameters and by embryo weight as an age proxy. All samples were scored using the criteria provided in **Supplementary file 1**.

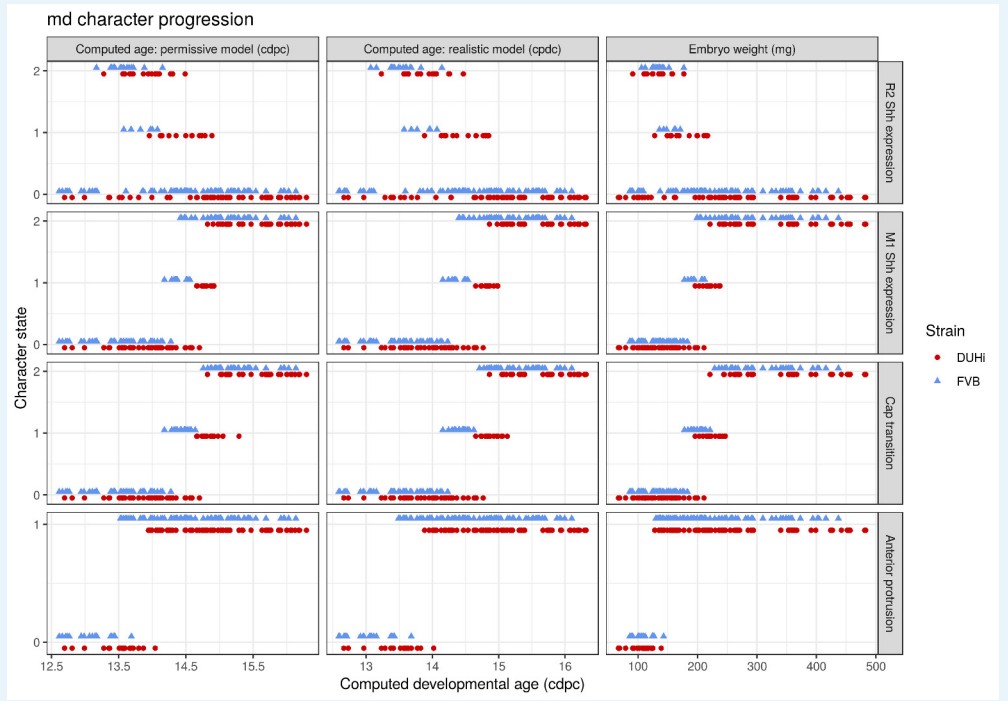

**Appendix 1—figure 3.** Comparative progression of scored characteristics in lower molar epithelia under different measures of age. Progression of four characters (R2 *Shh* expression, M1

*Shh* expression, cap transition and anterior protrusion of the dental epithelium) is depicted in samples of lower molars from two strains (FVB and DUHi). In order to allow the comparison of age measurements, the temporal axis is provided by computed embryonic age ($_c$dpc) under the realistic and permissive parameters and by embryo weight as an age proxy. All samples were scored using the criteria provided in *Supplementary file 1*.

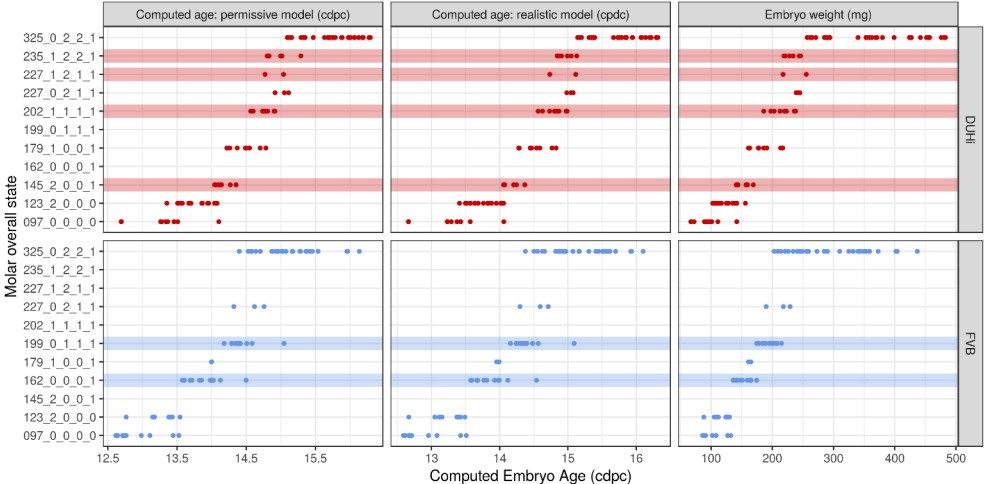

**Appendix 1—figure 4.** Total state of embryonic upper molars under different measures of age. Temporal distribution of developmental state of the developing upper molar, produced by combining a value for each of the four scores for a given sample, based on criteria from *Supplementary file 1*. Each of the developmental states observed are shown and are ordered according to the average embryonic weight of the samples within that group. Each state present is coloured according to whether it is found in DUHi only (red), in FVB only (blue), or in both DUHi and FVB (white) samples. In order to allow the comparison of age measurements, the temporal axis is provided by computed embryonic age ($_c$dpc) under the realistic and permissive parameters and by embryo weight as an age proxy.

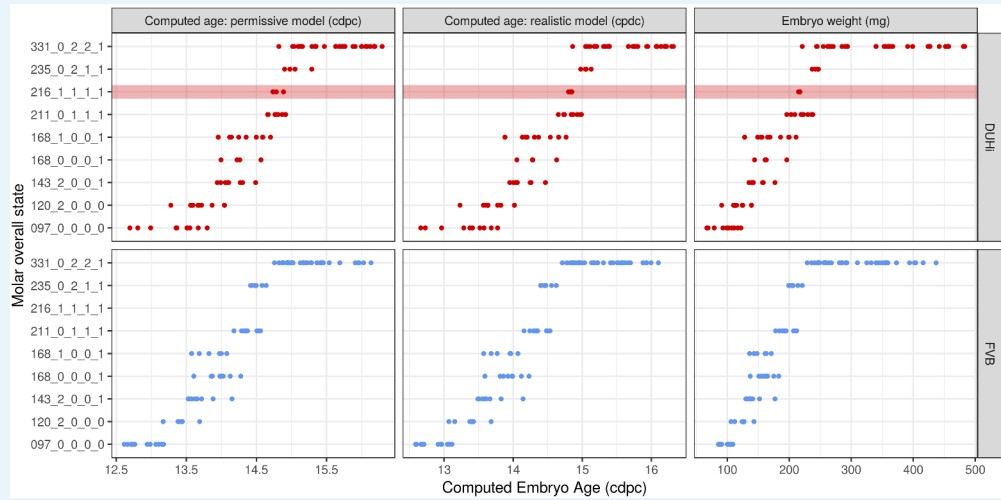

**Appendix 1—figure 5.** Developmental state of embryonic lower molars under different measures of age. Temporal distribution of developmental state of the developing lower molar, produced by combining a value for each of the four scores for a given sample, based on criteria from *Supplementary file 1*. Each of the developmental states observed are shown and are ordered according to the average embryonic weight of the samples within that group. Each state present is coloured according to whether it is found in DUHi only (red) or in both DUHi and FVB (grey) samples. In order to allow the comparison of age

measurements, the temporal axis is provided by computed embryonic age ($_c$dpc) under the realistic and permissive parameters and by embryo weight as an age proxy.

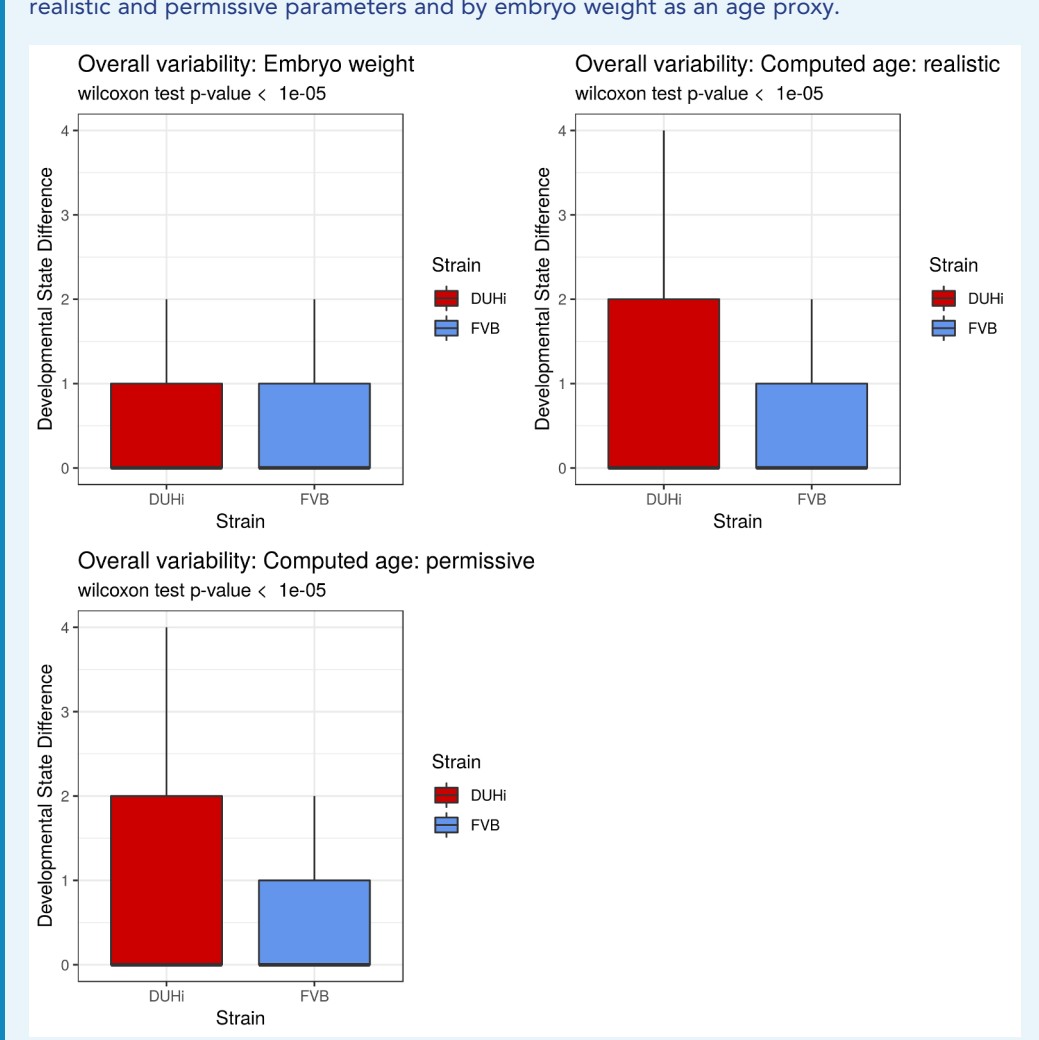

**Appendix 1—figure 6.** Developmental variability in the developing upper molars of DUHi and FVB mice embryos under different measures of age. Variability in the developing upper molar is represented as a boxplot of developmental state differences calculated for pairs of samples with less than 0.25 difference in computed embryonic age ($_c$dpc). Samples close in age are significantly more different in developmental state in DUHi versus FVB mice when the temporal axis is provided by computed embryonic age ($_c$dpc) under the realistic and permissive parameters and when embryo weight is used as an age proxy.

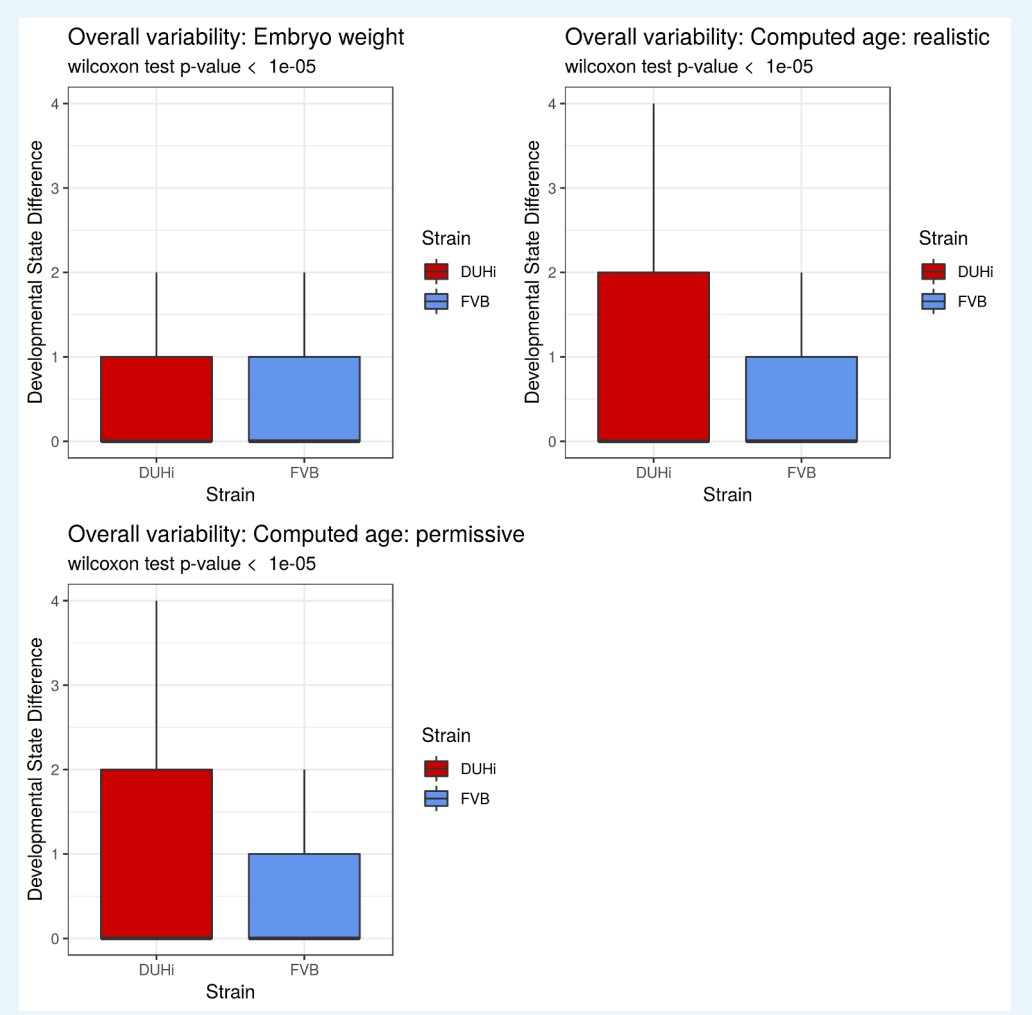

**Appendix 1—figure 7.** Developmental variation in the developing lower molars of DUHi and FVB mice embryos under different measures of age. Variation in the developing lower molar is represented as a boxplot of developmental state differences calculated for pairs of samples with less than 0.25 difference in computed embryonic age ($_c$dpc). Samples close in age are significantly more different in developmental state in DUHi versus FVB mice when the temporal axis is provided by computed embryonic age ($_c$dpc) under the realistic and permissive parameters.

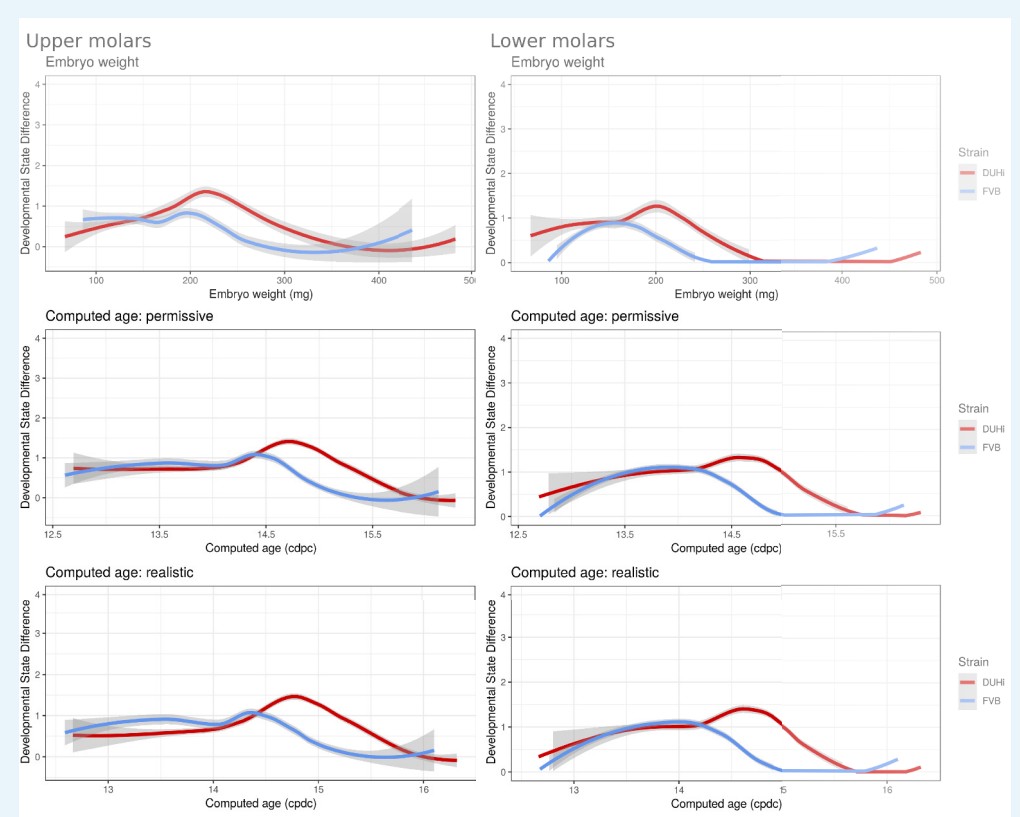

**Appendix 1—figure 8.** Developmental variation over time in the upper and lower molars of DUHi and FVB mice embryos under different measures of age. The mean developmental distance between nearby samples (embryonic age difference < 0.25 d) is plotted as the local regression line (LOESS smoothing) for both strains (standard deviation shown in grey). In order to allow the comparison of age measurements, the temporal axis is provided by computed embryonic age ($_c$dpc) under the realistic and permissive parameters and by embryo weight as an age proxy. Developmental variation is higher in DUHi under all measures of age, and peaks at time of R2 and M1 signaling center co-existence.

## RNAseq of DUHi and FVB M1 germs at cap stage

For each strain, we sampled 3 embryos with similar body weight (DUHi: 196, 219 and 239 mg) and FVB (195, 215 and 233 mg) for which we obtained both lower and upper tooth germ samples. The morphology of the tooth germs during the dissection process indicated that only the oldest DUHi embryo had just accomplished cap transition, while all three FVB embryos had just accomplished cap transition.

We retrieved 3619 DE (Differentially Expressed) genes (6 samples in FVB/in DUHi, jaw treated as a factor in DESeq2:~jaw + Strain). This high number of genes was partly due to the slight developmental time difference between FVB and DUHi samples around cap transition, with DUHi samples tending to be slightly younger than FVB samples (e.g. FVB-DUHi genes partly overlapped with genes differentially expressed across the 3 replicates ordered in time by embryo weight). In particular, enamel knot genes (e.g. *Shh, Dkk4, Slit1*) appeared upregulated in FVB samples that have just undergone cap transition, as compared with the youngest DUHi samples. Genes with GO terms (searched with Gorilla tool) associated with mitosis (mitotic cell cycle process, p-value $1.37 \cdot 10^{-11}$; chromosome segregation $1.74 \cdot 10^{-7}$) were enriched in FVB samples, whereas DUHi samples were enriched for amino-acid and carbohydrate metabolism genes (rRNA metabolic process p-value $4.76 \cdot 10^{-6}$, organonitrogen compound biosynthetic process p-value $9.82 \cdot 10^{-6}$) suggesting that FVB samples were enriched in mitotic cells whereas DUHi samples are enriched in G1/S/G2 growing cells. This might reflect the slight developmental delay in DUHi samples, if cap transition is associated with a sudden burst of mitosis. Another possibility is that this reflects metabolic differences between the two strains, since DUHi mice are large-sized mice.

First, we checked genes known to be involved in the formation of supernumerary tooth anterior to M1, resembling the premolar lost during mouse evolution. Among DE genes, we found two genes (*Spry2, Sostdc1*) whose mutation rescues premolar formation (***Figure 4—figure supplement 2***). *Spry4, Rsk2, Gas1, Lrp4, Eda, Edar, Fgf20*, whose knock-out or overexpression also rescues premolar formation were not differentially expressed. However, in differential expression analyses conducted independently on lower and upper jaw, *Gas1* was DE in lower jaw (but unbiased in upper jaw) and *Eda* was marginally significantly DE in the upper jaw only). Finally, treatments interfering with *Shh* also rescue premolar formation (***Cho et al., 2011***; ***Harjunmaa et al., 2012***). We note that several *Shh* pathway genes are downregulated in DUHi mice (e.g. *Shh, Gli1, Ptch1*). However, because *Shh* is rapidly and strongly upregulated at cap transition (***Prochazka et al., 2010***; this is also visible in our transcriptomes, see the ***Supplementary file 2***), it is hard to disentangle a true downregulation of the *Shh* pathway from an artifact caused by the developmental delay observed in DUHi mice.

Next, we examined the BMP4 and Wnt pathways in detail, based on a list of regulatory relationships published by ***O'Connell et al. (2012)***. This list contains a number of presumed targets activated or repressed by each pathway, based on downregulation and upregulation in mutants or upon treatments (e.g. treatment with BMP4, treatment against GSK3 to activate the Wnt pathway). We removed ambiguous genes from this list (i.e. that were described as activated or repressed depending on the tissue examined, epithelium or mesenchyme; e.g. dlx1). Then for each pathway, we examined genes of this list that were also differentially expressed between FVB and DUHi strains. Because the activation-inhibition (A-I) balance involves many genes with complex regulatory feedbacks within and between these two key pathways (and other pathways), we did not expect a change in the A-I balance to shift all targets genes in a consistent direction. Rather, we expected to find a different equilibrium, with genes changed in both directions, but that may nevertheless collectively indicate greater or instead weaker activation of these pathways in the DUHi mice. Genes were considered to be indicators of greater pathway activity in DUHi as compared to FVB, if they were both activated targets in the regulatory list and upregulated in DUHi strain, or both repressed target genes and downregulated in DUHi strain. Reciprocally, genes were considered to be

indicators of weaker pathway activity in DUHi if they were both activated targets in the regulatory list and downregulated in DUHi strain, or both repressed target genes and upregulated in DUHi strain. As mentioned in the main text, we found a marked bias for indicators of weaker BMP4 activity in DUHi samples. Although the low expression levels of some activated BMP4 targets in DUHi may partially be attributed to the developmental delay between strains (ie. BMP4 targets that are upregulated at the cap transition could have lower expression levels in the youngest DUHi samples because they have not yet accomplished the cap transition), repressed BMP4 targets exhibited high expression levels in DUHi without any obvious correlation with time differences in samples. Therefore, we believe that the low BMP4 activity in DUHi samples is not artefactually driven by the temporal difference in sampling.

