## [Decision Letter]

Thank you for submitting your article "Developmental variability drives mouse molar evolution along an evolutionary line of least resistance" for consideration by *eLife*. Your article has been reviewed by two peer reviewers, and the evaluation has been overseen by a Reviewing Editor and Diethard Tautz as the Senior Editor. The reviewers have opted to remain anonymous.

The reviewers have discussed the reviews with one another and the Reviewing Editor has drafted this decision to help you prepare a revised submission.

Summary:

Both reviewers, and myself, agree that this manuscript represents superb work that merits publication in *eLife*. As reviewer #2 states: This is superb work…shows at the fine scale how 'tinkering' of gene expression, signalling centres and physical spacing can add up to changes in variability and average morphology. Moreover, reviewer #1 states: The sensitivity to and care in examining variation in this paper is refreshing and makes it stand apart from many other such investigations in evolutionary developmental biology. Moreover, the points made about how variation across strains of mouse may influence our interpretations are very well stated and a welcome intervention. After discussion among the reviewers and myself, we have a few points we would like the authors to address, which are listed below.

Major points for revision:

1) The authors set out this study on the basis of investigating 'evolutionary lines of least resistance'. One of the main features that they describe as a 'line of least resistance' is the length/width ratio of the molars. Both reviewers and myself agree that we would like to ask for clarification/justification of the use of the term evolutionary lines of least resistance or perhaps a reframing of the problem (which should not take too much effort). We are not sure that the "evolutionary lines of least resistance is the best framing for this paper, but are willing to be convinced if the authors feel strongly. If not, we recommend reframing the paper.

2) Comment of reviewer #1: The interpretation of the developmental scoring presented in Supplementary file 1 and Figure 3 is not clear to me as presented in the paper. It strikes me that there are some dependencies within (at least one measure is ordered) and among (some trait states seem to depend on other trait states to some extent) the different traits. As such, it's not clear that there are actually 54 developmentally plausible and possible trait combinations simply because of the ways in which the order and timing of the development of the states play out. This merits some re-explanation to render it clearer to the reader and to fully explore the implications of trajectories in developmental space.

3) Comment of reviewer #1: While the activation/inhibition framework is popular in the study of molar teeth, it is also a bit vague and can encompass very nearly any kind of influence on the growth and development of molar teeth. It would be helpful if each of the proposed mechanisms affecting activation and/or inhibition were more clearly connected to their precise roles. This is particularly an issue with the gene expression results because there is not an internal control to account for more systemic differences between strains. That, and it is becoming clear that much of the variation in mammalian molar form found within species does not neatly conform to an activation/inhibition model of tooth development. Having a concrete example of something that is not an activation/inhibition factor might go a long way toward clarifying the discussion of these conclusions.

---

## [Author Response]

Major points for revision:1) The authors set out this study on the basis of investigating 'evolutionary lines of least resistance'. One of the main features that they describe as a 'line of least resistance' is the length/width ratio of the molars. Both reviewers and myself agree that we would like to ask for clarification/justification of the use of the term evolutionary lines of least resistance or perhaps a reframing of the problem (which should not take too much effort). We are not sure that the "evolutionary lines of least resistance is the best framing for this paper, but are willing to be convinced if the authors feel strongly. If not, we recommend reframing the paper.

We believe that the notion of line of least resistance, as far as we understand it, is appropriate to our study. However, we are fully ready to consider the reasons “why the evolutionary lines of least resistance would not be the best framing for this paper”, and reframe the paper accordingly. Yet, we did not clearly get the reasons why it is so. Maybe this relies on a misunderstanding about what variation is considered; actually, this is not merely a trivial matter of a length/width ratio. Maybe we also use sometimes the terminology “lines of least resistance” for referring to a direction of preferred variance (which, however, could act as LLR)? Maybe a less specific terminology would be preferred (i.e. simply developmental constraints channeling variation and phenotypic evolution)? These are the possibilities that we envisioned and we replied in detail below. We remain ready to consider further changes, if the issue is more clearly explicated.

We have tried to better explain why we consider to have one main direction of variance, supported by two intermingled geometrical components: 1) one regards the general lower and upper molar morphology: the main direction of variation not only involves the general length/width ratio, but overall shape: narrow shape with narrow cusps *versus* broad shape with broad cusps. These extreme morphologies are associated with marked dietary preferences, suggesting that this type of variation could be adaptive. However, between these extremes, variation is continuous in nature and the same kind of variation is found at all phylogenetic levels, from intra-population to interspecies level (extant of variation is of course different). Such a continuum, and especially the fine intra-population variation, is more likely to be shaped by developmental properties than by adaptation alone. The alignment of the main phenotypic variation across the taxonomic scale therefore suggests that murine molars evolve of along a line of least resistance, with adaptation occurring along the line imposed by developmental properties. 2) The second type of variation is specific to the upper molar, which shows extra lengthening often associated with an anterior supplementary cusp. Again, this variation is found at all phylogenetic levels, meeting the definition of a direction of primary variance channeling the evolution of populations and species, i.e. acting as a line of least resistance.

We are conscious that introducing this vocabulary may not be fully necessary, the notion of “developmental constraint channeling evolution” could be sufficient and more readily accessible to any reader. On the other hand, we feel that using this vocabulary, probably more familiar to morphometricians and geneticists, is an opportunity to bring them to an experimental evolutionary developmental biology study.

To accommodate this, we tried to reduce the emphasis on “evolutionary line of least resistance”, by removing it from the title, and rephrasing some parts. We have also further attempted to separate the variation in tooth geometry from its evolutionary role as a line of least resistance.

2) Comment of reviewer #1: The interpretation of the developmental scoring presented in Supplementary file 1 and Figure 3 is not clear to me as presented in the paper. It strikes me that there are some dependencies within (at least one measure is ordered) and among (some trait states seem to depend on other trait states to some extent) the different traits. As such, it's not clear that there are actually 54 developmentally plausible and possible trait combinations simply because of the ways in which the order and timing of the development of the states play out. This merits some re-explanation to render it clearer to the reader and to fully explore the implications of trajectories in developmental space.

We acknowledge that for 3 out of 4 traits, the states are ordered, reflecting the normal unfolding of development.

This is true for the two morphological traits (anterior protrusion, cap transition) and one molecular trait (M1 Shh expression). A bud (0) should always be seen before a cap (2). M1 Shh expression starts from 0: no expression to 1: round zone to 2: oval zone. This reflects the maturation of the M1 signaling center.

R2 expression however is not oriented: 0: no expression, 1: weak, 2: strong. We put no a prioriassumptions on it, as almost all transitions seem possible 0->1->2, 2->1->0, but also 0- >2 or 2->0. Here, combining this criterion with “anterior protrusion” and “cap transition” provides the orientation. No expression or weak expression with flat dental epithelium characterizes early development; No expression or weak expression with conspicuous anterior protuberance and/or early or late signs of cap transition characterizes later development.

Although there are 54 mathematically possible trait combinations, we fully agree that less combinations are developmentally plausible, and again this dependence between traits reflects the conserved unfolding of “normal” development. For example, transitioning from a well-developed cap to a bud seems irrelevant, because normal tooth development proceeds from bud to cap stage. Yet it remains conceivable in mutant situations, where morphogenesis would be stopped and the tissue would start regress. Combining no protuberance in R2 zone with a cap transition is unexpected, because R2 develops before M1. Combining large M1 *Shh* expression and no cap transition is also unexpected, because this large expression marks the presence of a PEK and PEK drives cap transition. Yet, again in mutant mice, things may happen differently, because the normal rules of development may be broken (e.g. defective PEK despite normal *Shh* expression). This emphasizes that the actual occupied proportion of developmental space and the trajectories through this space are properties of a given genotype with a given development. In practice, most combinations are shared by the two strains, as well as time ordering of these combinations. A few combinations are strain-specific. Moreover, the time spent in a given trait combination can be strain-specific.

We have tried to make this clear in the text.

3) Comment of reviewer #1: While the activation/inhibition framework is popular in the study of molar teeth, it is also a bit vague and can encompass very nearly any kind of influence on the growth and development of molar teeth. It would be helpful if each of the proposed mechanisms affecting activation and/or inhibition were more clearly connected to their precise roles. This is particularly an issue with the gene expression results because there is not an internal control to account for more systemic differences between strains. That, and it is becoming clear that much of the variation in mammalian molar form found within species does not neatly conform to an activation/inhibition model of tooth development. Having a concrete example of something that is not an activation/inhibition factor might go a long way toward clarifying the discussion of these conclusions.

We fully agree with the reviewer that the A/I framework in its original definition (Kavanagh et al.) is quite vague. Our recent attempt (Sadier et al., 2019) to model the sequential formation of signalling centers in the molar field somehow teases apart different aspects (e.g. distinguishing posterior growth rate, activation and inhibition in the epithelium, epithelium maturation, mesenchymal influence…), but we did not identify the underlying molecular framework, as was done in other systems (i.e. hair or feather). Because the formation of M1 was long mistaken with the formation of vestigial buds and the antero-posterior dimension was long neglected, most mouse mutant require reinterpretation to achieve a molecular understanding of signalling center sequential formation. At the moment, BMP4 and Wnt pathways (at least different Wnt pathways are involved in epithelium and mesenchyme, with opposite effects on tooth formation) are clearly major actors in tooth formation, but their exact roles, in promoting activation-inhibition sensu *stricto*, posterior growth, tissue maturation (including through mesenchymal condensation) remain to be defined. Other pathways are clearly involved (notably FGF and Activin; components of both differ in expression between FVB and DUHi) but their exact role is unclear as well.

Unfortunately, our model from the Sadier et al., 2019 paper is fitted for lower molar, rather than for upper molar formation. However, we can get some clues from the sensitivity analysis performed in the supplementary material (Figure 2—figure supplement 1, 2 and 3). In this analysis, the triggering of 3 different parameters increased the distance between R2 and M1 signaling centers, as seen between FVB and DUHi mice:

– Changing A/I parameters sensu *stricto* (changing the activator/inhibitor balance is achieved in our model by changing the rate of auto-inhibition of the inhibitor, this leads to more distant signaling centers and rescued R2).

– Changing the rate of production of the mesenchymal signal that primes the tissue for activation-inhibition mechanisms (decreasing it leads to more distant signalling centers, but without R2 rescue; increasing it also leads to more distant signalling centers and rescued R2. Thus this parameter has very context-dependent effects).

– Changing posterior growth (increasing it leads to more distant signaling centers, and could lead to R2 rescue). This was an obvious possible difference between DUHi and FVB mice, as DUHi embryos are larger than FVB embryos for a similar developmental stage and increase in mouse size is repeatedly associated with anterior lengthening of upper molars. Our measurements found no significant difference in DUHi vs. FVB total length of the dental lamina. Therefore, it seems that we can rule out a simple growth effect. We do not totally exclude that a systemic difference between DUHi and FVB mice could change the expansion rate of the region competent to form teeth, without affecting the total lamina length. This would merit further work.

The BMP4 and Wnt pathways are possible molecular effectors both for activation-inhibition sensu *stricto* and for the production of the mesenchymal signal, as formalized in our model. In both cases, our sensitivity analysis suggests that decreasing their activity could result in more distant signaling centers as observed in DUHi.

We have introduced most of these considerations in the Results and Discussion, and revised the text to avoid using “activation-inhibition balance” in an undetermined manner. We do not see how we could elaborate much further, based on current knowledge. Working on the molecular basis of signaling center formation and the differences between FVB and DUHi mice is something we are pursuing in the lab, but it is out of focus of the present study.